# Timing mechanism of sexually dimorphic nervous system differentiation

**Laura Pereira[1]\*, Florian Aeschimann[2,3], Chen Wang[1], Hannah Lawson[4], Esther Serrano-Saiz[1], Douglas S Portman[4,5], Helge Großhans[2,3], Oliver Hobert[1]\***

[1]Department of Biological Sciences, Howard Hughes Medical Institute, Columbia University, New York, United States; [2]Friedrich Miescher Institute for Biomedical Research, Basel, Switzerland; [3]University of Basel, Basel, Switzerland; [4]Department of Biology, University of Rochester, Rochester, United States; [5]DelMonte Institute for Neuroscience, Department of Biomedical Genetics, University of Rochester, New York, United States

**Abstract** The molecular mechanisms that control the timing of sexual differentiation in the brain are poorly understood. We found that the timing of sexually dimorphic differentiation of postmitotic, sex-shared neurons in the nervous system of the *Caenorhabditis elegans* male is controlled by the temporally regulated miRNA *let-7* and its target *lin-41*, a translational regulator. *lin-41* acts through *lin-29a,* an isoform of a conserved Zn finger transcription factor, expressed in a subset of sex-shared neurons only in the male. Ectopic *lin-29a* is sufficient to impose male-specific features at earlier stages of development and in the opposite sex. The temporal, sexual and spatial specificity of *lin-29a* expression is controlled intersectionally through the *lin-28/let-7/lin-41* heterochronic pathway, sex chromosome configuration and neuron-type-specific terminal selector transcription factors. Two Doublesex-like transcription factors represent additional sex- and neuron-type specific targets of LIN-41 and are regulated in a similar intersectional manner.

DOI: https://doi.org/10.7554/eLife.42078.001

\*For correspondence:
pereira.lau@gmail.com (LP);
or38@columbia.edu (OH)

**Competing interest:** See
page 26

**Reviewing editor:** Piali
Sengupta, Brandeis University,
United States

## Introduction

The nervous systems of males and females of many animal species display anatomical and functional differences (*Jazin and Cahill, 2010*; *McCarthy and Arnold, 2011*; *Portman, 2017*; *Yang and Shah, 2014*). Sex-specific maturation events in the brain are typically triggered at a specific window of postnatal life. Sexual maturation in mammals, called puberty, represents a key developmental transition and is accompanied by substantial sexually dimorphic changes in the brain (*Avendaño et al., 2017*; *Juraska and Willing, 2017*; *Sisk and Foster, 2004*). The onset of puberty in mammals is characterized by the activation of neurons in the hypothalamus that produce hormonal signals (*Avendaño et al., 2017*). However, the mechanisms that control the activation of these neurons and hence the timing of pubertal onset are poorly understood.

Like in other animals, most of the sex-specific features of the nervous system of the nematode *Caenorhabditis elegans* are also established during postembryonic development (*Barr et al., 2018*; *Sulston et al., 1980*; *Sulston and Horvitz, 1977*). Sexual dimorphisms in the *C. elegans* nervous system fall into two broad types: (1) Some neurons are present only in one sex but not the opposite sex (*Sulston et al., 1980*; *Sulston et al., 1983*). These sex-specific neurons are directly involved in distinct aspects of sex-specific mating and reproductive behavior (*Emmons, 2018*). (2) Some embryonically born, postmitotic neurons that are present in both sexes ('sex-shared' neurons) can adopt sex-specific features during sexual maturation (*Portman, 2017*). Sexually dimorphic features of sex-shared neurons include sex-specific synaptic wiring patterns (*Jarrell et al., 2012*) and sex-specific expression of signaling molecules, such as neurotransmitters, neuropeptides and chemoreceptors

**eLife digest** In most adult animals, male and female brains are slightly different. For example, in the nematode worm *Caenorhabditis elegans*, certain neurons exist in one sex but not the other. Nerve cells that are shared in both sexes may also activate different genes, or form different connections in males and females. Most of these differences – which ultimately give rise to sex-specific behaviors – emerge during a period of development called sexual maturation. Yet, the mechanisms that control when sexual differentiation takes place in the brain are largely unknown.

To investigate this, Pereira et al. set out to determine how sex differences arise in the nervous system of *C. elegans*, a small animal with two sexes, male and hermaphrodite. In particular, Pereira et al. wanted to know which genes cause certain neurons that are present in both sexes to switch to the male-specific form when the worm gets old enough.

The experiments revealed that a genetic pathway formed of three genes, *let-7*, *lin-28* and *lin-41*, controls when sexual maturation takes place throughout the worm nervous system. When the worm is young, *lin-41* is active and represses a gene called *lin-29A*. As the animal reaches maturity, *let-7* 'switches off' *lin-41*, and *lin-29A* gets activated in a subset of neurons. These brain cells then turn on male-specific genes and acquire a shape only found in males. The anatomy of male mutant worms that lack *lin-29A* is normal, but the animals show features found in hermaphrodites, for example in the way they crawl across a dish. This shows that activating *lin-29A* may also trigger male-specific behaviors.

Switching on sex-specific neuronal circuits at the correct time is essential for animals to develop correctly. The *lin-7* and *let-28* genes also control when sexual maturation takes place in mammals, so studying these genes in *C. elegans* could help to understand how male and female brains are shaped during development in other species, and why some diseases affect the sexes differently.

DOI: https://doi.org/10.7554/eLife.42078.002

(*Hilbert and Kim, 2017*; *Pereira et al., 2015*; *Ryan et al., 2014*; *Serrano-Saiz et al., 2017a*; *Serrano-Saiz et al., 2017b*). For example, the AIM interneuron class is glutamatergic in juvenile, pre-sexual maturation stages of both sexes, but during sexual maturation will turn off glutamatergic identity and turn on cholinergic neurotransmitter identity specifically in males (*Pereira et al., 2015*). In other cases, sexually dimorphic features emerge from a 'mixed' juvenile state. For example, juvenile phasmid sensory neurons innervate several distinct interneuron targets in both sexes before sexual maturation, but during sexual maturation specific subsets of these connections get pruned in a sex-specific manner (*Oren-Suissa et al., 2016*).

Most if not all of these sexually dimorphic features of the *C. elegans* nervous system terminally differentiate during the fourth larval stage (*Desai et al., 1988*; *Oren-Suissa et al., 2016*; *Pereira et al., 2015*; *Serrano-Saiz et al., 2017a*; *Sulston and Horvitz, 1977*). The molecular mechanisms that control the timing of sexual maturation in postmitotic neurons have largely remained obscure. Candidate regulators of sex-specific developmental timing events are so-called heterochronic genes, first identified in *C. elegans* (*Ambros and Horvitz, 1984*). Heterochronies refer to offsets of the relative timing of somatic developmental events versus gonadal developmental events (*Slack and Ruvkun, 1997*). In heterochronic mutants, the timing of cellular cleavage patterns in postembryonic skin cell lineages is disrupted in a highly stereotyped manner; in some mutants, lineage patterning events normally occurring late will occur earlier ('precocious mutants') while in other mutants specific cell lineage patterning events will be delayed or never occur (*Ambros, 1989*; *Ambros and Horvitz, 1984*; *Slack and Ruvkun, 1997*). Molecular analysis of these heterochronic mutants revealed a gene regulatory pathway, which contains several deeply conserved post-transcriptional regulatory factors at its core, including the miRNA *let-7*, the LIN-28 protein, a negative regulator of *let-7* processing, and the RNA-binding, posttranscriptional regulator LIN-41, a key target of the *let-7* miRNA (*Figure 1A*) (*Ambros, 1989*; *Ecsedi et al., 2015*; *Moss et al., 1997*; *Reinhart et al., 2000*; *Slack et al., 2000*). While these regulatory factors are well known to control cell division patterns in the ectoderm, their role in postmitotic cell types is much less well defined (*Del Rio-Albrechtsen et al., 2006*; *Hallam and Jin, 1998*; *Howell et al., 2015*; *Olsson-Carter and Slack, 2010*; *Zou et al., 2013*).

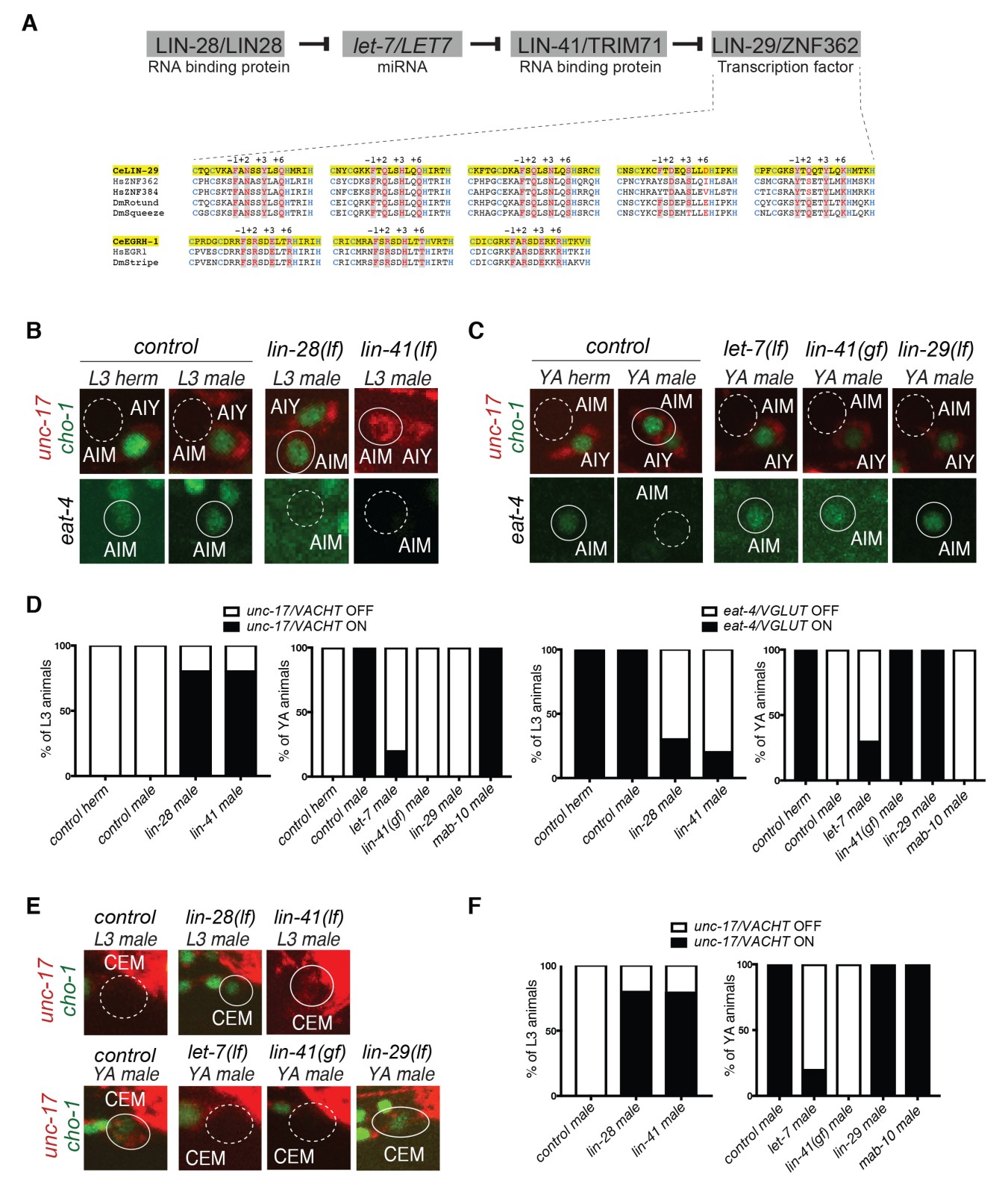

**Figure 1.** Heterochronic pathway effects on AIM and CEM neuronal differentiation during sexual maturation. (**A**) A simplified schematic of the heterochronic pathway, deduced from the genetic analysis of hypodermal cell lineage division events. The LIN-28 RNA binding protein prevents the maturation of miRNA *let-7*. When LIN-28 protein levels go down at the end of the second larval stage (L2), mature miRNA *let-7* levels rise and bind to the 3'UTR of *let-7* targets. One *let-7* target is the RNA-binding protein LIN-41, whose levels are downregulated at the onset of sexual maturation (L4).

*Figure 1 continued on next page*

*Figure 1 continued*

This allows for its negative regulator, the zinc-finger transcription factor *lin-29,* to be expressed at the L4 stage to induce the larval to adult transition. Sequence homology alignment of the five Zn-finger domains shows that LIN-29 is an orthologue of human ZNF362 and ZNF384, while EGRH-1 is an orthologue of human EGR. The Zn fingers of worm, fly and human are shown. Zn coordinating Cys and His residues are colored in blue and DNA-contacting residues (at position −1,+2,+3 and+6 of the helixes preceding the His residues) are colored in red. Conservation is indicated in grey shading. (B) AIM neuronal differentiation is precocious in *lin-28(n719lf)* and *lin-41(bch28lf). unc-17/VACHT* (*ot907*) and *cho-1/CHT* (*otIs534*) cholinergic reporter expression (cytoplasmic for *unc-17* and nuclear for *cho-1*) is not observed in AIM in control animals at the L3 stage (dotted circles, top panels). *lin-28(n719lf)* and *lin-41(bch28lf)* mutant males show precocious cholinergic gene expression in AIM at the L3 stage (top panels). *eat-4/VGLUT* (*otIs388*) glutamatergic reporter expression is observed in AIM in control animals at the L3 stage (circled in white, bottom panels). *lin-28(n719lf)* and *lin-41 (bch28lf)* mutant males show precocious loss of AIM glutamatergic identity (dotted circles, bottom panels). The AIY cholinergic neuron is located next to AIM and is used as a positional reference. Solid circles indicate expression of the reporter, stippled circle indicates loss of expression. (C) Male-specific AIM differentiation is blocked in *let-7(n2853ts), lin-41(xe8gf)* and *lin-29a/b(n333lf)* mutants. *unc-17/VACHT* (*ot907*) and *cho-1/CHT* (*otIs354*) cholinergic reporter expression (cytoplasmic for *unc-17* and nuclear for *cho-1*) is observed in young adult (YA) control males but not in AIM neurons in hermaphrodites nor *let-7(n2853ts), lin-41(xe8gf)* and *lin-29a/b(n333lf)* mutant males (dotted circles, top panels). *eat-4/VGLUT* (*otIs388*) glutamatergic reporter expression is observed in hermaphrodites and in *let-7(n2853ts), lin-41(xe8gf)* and *lin-29a/b(n333lf)* mutant males, where it fails to be downregulated (circled in white, bottom panels). L1 *let-7(n2853ts)* animals were shifted to the restrictive temperature (25°C) and imaged after 48hs. The incomplete penetrance of the *let-7* mutants is likely because the allele used, *n2853,* a point mutation in the miRNA seed region, is hypomorphic. The AIY cholinergic neuron is located next to AIM and is used as a positional reference. (D) Quantification for the AIM neurotransmitter switch in heterochronic mutants *lin-28(n719lf), lin-41(bch28lf), let-7(n2853ts), lin-41(xe8gf), lin-29a/b(n333lf)* and *mab-10(xe44)* (n = 15). Expression of *unc-17/ VACHT* (*ot907*) and *eat-4/VGLUT* (*otIs388*) was quantified as ON or OFF. (E) CEM neuronal differentiation defects in heterochronic mutants. CEM cholinergic gene expression is precocious in *lin-28(n719lf)* and *lin-41(bch28lf)* males compared to control. *unc-17/VACHT* (*ot907*) and *cho-1/CHT* (*otIs354*) cholinergic reporter expression is not observed in CEM neurons in wild-type males at the L3 stage (dotted circle). *lin-28(n719lf)* and *lin-41 (bch28lf)* males show precocious cholinergic gene expression in CEM neurons at the L3 stage (circled in white, top panels). CEM cholinergic gene expression is lost in *let-7(n2853ts)* and *lin-41(xe8gf)* mutants (dotted circles). *lin-29a/b(n333lf)* mutants showed no defect in cholinergic gene expression in CEM neurons (circled in white). L1 *let-7(n2853ts)* animals were shifted to the restrictive temperature (25°C) and imaged after 48hs. The incomplete penetrance of the *let-7* mutants is likely because the allele used, *n2853,* a point mutation in the miRNA seed region, is hypomorphic. (F) Quantification for CEM cholinergic gene expression in heterochronic mutants *lin-28(n719lf), lin-41(bch28lf), let-7(n2853ts), lin-41(xe8gf), lin-29a/b(n333lf)* and *mab-10 (xe44)* (n = 15). Expression of *unc-17/VACHT* (*ot907*) was quantified as ON or OFF.

DOI: https://doi.org/10.7554/eLife.42078.003

The following figure supplements are available for figure 1:

**Figure supplement 1.** Expression pattern for the endogenously tagged *unc-17/VACHT* allele during development.
DOI: https://doi.org/10.7554/eLife.42078.004
**Figure supplement 2.** AIM and CEM neuronal differentiation does not require the gonad nor the germline.
DOI: https://doi.org/10.7554/eLife.42078.005

Within the nervous system, the impact of heterochronic genes on sex-specific differentiation events during sexual maturation at the juvenile to adult transition has been poorly defined. Such a role is not an obvious matter because heterochronic genes do not control timing in all tissues. For example, heterochronic genes are not involved in controlling the timing of gonadal maturation (*Ambros and Horvitz, 1984*; *Euling and Ambros, 1996*). We show here that the phylogenetically conserved core heterochronic pathway genes *lin-28, let-7* and *lin-41* control the timing of maturation of sexual dimorphisms in different neuron classes. We show that these broadly expressed timing genes act through distinct cell-specific effector modules, among them a specific isoform of the *lin-29* Zn finger transcription factor and the two DM domain (DMD) containing transcription factors *mab-3* and *dmd-3*. These temporally regulated effector genes are expressed in a sexually dimorphic manner in distinct sets of sex-shared neurons to control specific aspects of sexually dimorphic features and functions. Remarkably, genome-wide association studies have linked one component of the heterochronic pathway to aberrant onset of puberty in humans (*Faunes and Larraín, 2016*). Our findings therefore suggest the emergence of a phylogenetically conserved mechanism to control the timing of sexual maturation.

## Results

### Effects of the heterochronic *lin-28/let-7/lin-41* pathway on neuron-specific sexual maturation

As an entry point to identify the molecular mechanisms required for the correct timing of nervous system sexual maturation, we used the previously described neurotransmitter switch of the AIM interneurons (*Figure 1B–D*), a sex-shared neuron class that is required for mate searching behavior (*Barrios et al., 2012*; *Pereira et al., 2015*). A fosmid-based reporter of the *eat-4/VGLUT* vesicular glutamate transporter is expressed during juvenile stages in the AIM interneuron of both sexes. At the onset of sexual maturation, during the fourth larval stage, AIM neurons of the male turn off the glutamatergic gene reporter and turn on expression of fosmid-based reporters for the cholinergic genes *unc-17/VACHT* and *cho-1/CHT* (*Pereira et al., 2015*). The acquisition of AIM cholinergic identity during sexual maturation only in males was confirmed by analyzing the expression of the cholinergic *unc-17/VACHT* locus, tagged with *mKate2* using CRISPR/Cas9 genome engineering (*Figure 1—figure supplement 1*).

Sexual maturation in many animal species is controlled by gonadal hormones (*Avendaño et al., 2017*). In *C. elegans,* the gonad has been shown to be required for the establishment of sex-specific adult behaviors (*Arantes-Oliveira et al., 2002*; *Fujiwara et al., 2016*; *Kleemann et al., 2008*; *Lipton et al., 2004*). Therefore, we first examined whether signals from the germline or gonad may affect the AIM neurotransmitter switch. We found that neither genetic removal of the germ cells nor laser ablation of gonadal precursor cells (which results in the loss of both gonad and germline) affects the execution of the AIM neurotransmitter switch (*Figure 1—figure supplement 2*).

Temporal control of cell lineage decisions during early larval stages in *C. elegans* has been shown to depend on the activity of the evolutionary conserved heterochronic pathway (*Ambros and Horvitz, 1984*; *Moss and Romer-Seibert, 2014*; *Slack and Ruvkun, 1997*). To test if members of the heterochronic pathway were required for the correct timing of nervous system differentiation during sexual maturation, we examined the impact of heterochronic mutants on the male-specific AIM neurotransmitter switch. We observed no defects in the timing of cholinergic gene battery induction in L4 in *lin-4* null mutant animals which controls the timing of L2-specific aspects of cellular patterning (*Ambros and Horvitz, 1984*) (data not shown). However, we observed a premature onset of both *unc-17/VACHT* and *cho-1/CHT* expressions in AIM neurons of *lin-28* mutant males (*Figure 1B,D*), mirroring the precocious cellular cleavage defects of *lin-28* in epidermal tissues (*Ambros and Horvitz, 1984*). Matching the precocious induction of cholinergic identity, *lin-28* mutant males also showed a precocious downregulation of the glutamatergic marker *eat-4/VGLUT* in the AIM interneurons (*Figure 1B,D*).

We also found that in males carrying a loss-of-function mutation in the miRNA *let-7*, whose processing is negatively controlled by *lin-28* (*Lehrbach et al., 2009*), the neurotransmitter switch of the AIM interneurons is eliminated, that is, male AIM neurons remain glutamatergic (*Figure 1C–D*). Since *let-7* is a repressor of gene expression, the loss of the neurotransmitter switch suggests that *let-7* may normally act to repress a factor that prevents the neurotransmitter switch. We found that *lin-41*, the first of many *let-7* targets to be identified (*Slack et al., 2000*) encodes this repressor factor. In *lin-41* loss-of-function mutants (*bch28*), the neurotransmitter switch was prematurely executed, while in *lin-41* gain-of-function mutants (*xe8*), in which *let-7* is unable to downregulate *lin-41* activity due to a deletion of the *let-7*-binding sites in the *lin-41* 3'UTR (*Ecsedi et al., 2015*), the neurotransmitter switch of AIM was abrogated and AIM neurons remained glutamatergic (*Figure 1B,D*).

The heterochronic pathway genes also controlled the onset of differentiation of the male-specific CEM sensory neurons. Even though these neurons are already born in the embryo (and removed by apoptosis in hermaphrodites [*Sulston et al., 1983*]), the CEM neurons only fully differentiate at the fourth larval stage (*Pereira et al., 2015*). By analyzing the expression of cholinergic reporters, we found that this L4-specific maturation event is also precisely timed by the heterochronic pathway genes. In *lin-28* mutants, CEM neurons prematurely expressed cholinergic genes while in *let-7* mutants CEM neurons' terminal differentiation was blocked (*Figure 1E,F*). As we had seen for the AIM neurotransmitter switch, *lin-41* mutants showed precocious cholinergic gene expression in CEM neurons and *lin-41* gain-of-function mutants (in which *lin-41* expression fails to be downregulated) also fail to show CEM cholinergic gene expression during sexual maturation suggesting that *let-7* miRNA acts through regulating *lin-41* for the correct timing of this process (*Figure 1E,F*). However,

as in the case of the AIM neurotransmitter switch, *lin-4* mutants show no defect in the timing of CEM differentiation (data not shown). In conclusion, mutant analysis of the heterochronic pathway genes showed that the *lin-28/let-7/lin-41* regulatory pathway is required for the correct timing and for the acquisition of male-specific features of two distinct neuron classes; however, this regulatory cassette may couple to upstream regulatory inputs other than *lin-4*.

## The *lin-29* Zn finger transcription factor is the relevant target of *lin-41* in controlling the AIM neurotransmitter switch

*lin-41* encodes a versatile regulatory protein with RING and NHL domains, involved in both post-transcriptional RNA regulation (including translation) and protein ubiquitination (*Ecsedi and Großhans, 2013*). To ask which of the many previously described effectors of LIN-41 (*Aeschimann et al., 2017*; *Ecsedi and Großhans, 2013*) are relevant in the context of the AIM neurotransmitter switch, we turned to the *lin-29* Zn finger transcription factor, whose expression is translationally inhibited by *lin-41* in the context of timing of skin cell proliferation (*Aeschimann et al., 2017*; *Ambros and Horvitz, 1984*; *Rougvie and Ambros, 1995*; *Slack et al., 2000*)(*Figure 1A*). Orthology analysis with several tools (*Kim et al., 2018*; *Wang et al., 2017*) as well as a detailed analysis of predicted DNA binding features suggests that LIN-29 is an ortholog of the uncharacterized vertebrate ZNF362 and ZNF384 proteins (*Figure 1A*). During the larva-to-adult transition, *let-7* inhibition of *lin-41* derepresses *lin-29* expression and induces adult gene expression programs in the epidermis. Consistent with *lin-29* being a functional effector of *lin-41* in the AIM interneurons, we find that the AIM neurotransmitter switch is entirely lost in *lin-29* null mutants (*Figure 1C,D*). In contrast, we found that the onset of differentiation of the male-specific CEM neurons in the L4 stage, as measured by the induction of cholinergic identity, is unaffected in *lin-29* null mutants (*Figure 1E,F*). These findings indicate that *lin-41* controls distinct effector genes in different neuron types.

The transcriptional co-factor *mab-10/NAB* has been shown to act with *lin-29* to control epidermal cell differentiation during the larval to adult transition (*Harris and Horvitz, 2011*). Moreover, biochemical studies revealed that the *mab-10* transcript, like *lin-29*, is a direct target of the LIN-41 protein (*Aeschimann et al., 2017*). We confirmed this notion by CRISPR/Cas9-mediated tagging of the *mab-10* locus with *mCherry*, finding that ubiquitous *mab-10* expression initiates at the L4 stage (*Figure 2A*). However, analyzing the effects of a newly generated *mab-10* null allele, we found that in the absence of *mab-10*, the AIM neurotransmitter switch and CEM cholinergic gene expression occurred normally (*Figure 1D,F*). Hence, while *mab-10* functions with *lin-29* in seam cell differentiation, it is not required for *lin-29* function in AIM interneurons in males.

## Transition from ubiquitous temporal regulators (*lin-28/let-7/lin-41*) to a sex- and neuron-type specific regulator (*lin-29*)

To address in which cells the heterochronic pathway acts to control the AIM neurotransmitter switch, we first examined the expression pattern of *lin-28*, *let-7*, *lin-41* and *lin-29* in the nervous system. While expression of these genes in other cells, specifically, the epidermis, has previously been examined in detail (*Moss et al., 1997*; *Rougvie, 2001*; *Slack et al., 2000*), much less is known about their neuronal expression. We analyzed heterochronic gene expression relative to either a panneuronally expressed RFP marker or a cholinergic RFP marker (cholinergic labeling of the AIM neuron allowed us to correlate the timing of heterochronic gene expression relative to the AIM neurotransmitter switch). Consistent with previous reporter gene analysis (*Moss et al., 1997*), we found that a fosmid-based reporter construct of the *lin-28* gene is expressed very broadly in the nervous system in a sexually non-dimorphic manner and downregulated by the end of the L2 stage (*Figure 2B*).

To analyze *lin-41* expression, we used an endogenously *gfp* tagged *lin-41* allele (*Spike et al., 2014*). This *lin-41* reporter allele showed broad expression in the nervous system from the L1 until the L4 stage in both sexes, when it becomes downregulated (*Figure 2C*). Previous work showed that *lin-41* downregulation in hypodermal tissue depends on *let-7* (*Ecsedi et al., 2015*; *Slack et al., 2000*) and to corroborate this notion in the context of the nervous system, we analyzed *let-7* activity using an 'activity sensor' construct in which the *lin-41* promoter is fused to *gfp* and to the *let-7* responsive 3'UTR of the *lin-41* locus. This reporter shows ubiquitous expression, including in the nervous system, from the L1 stage to L4, when GFP levels start to drop and disappear from most tissues, including neurons, by the adult stage (*Figure 2D*). A control construct carrying the *unc-54* 3'

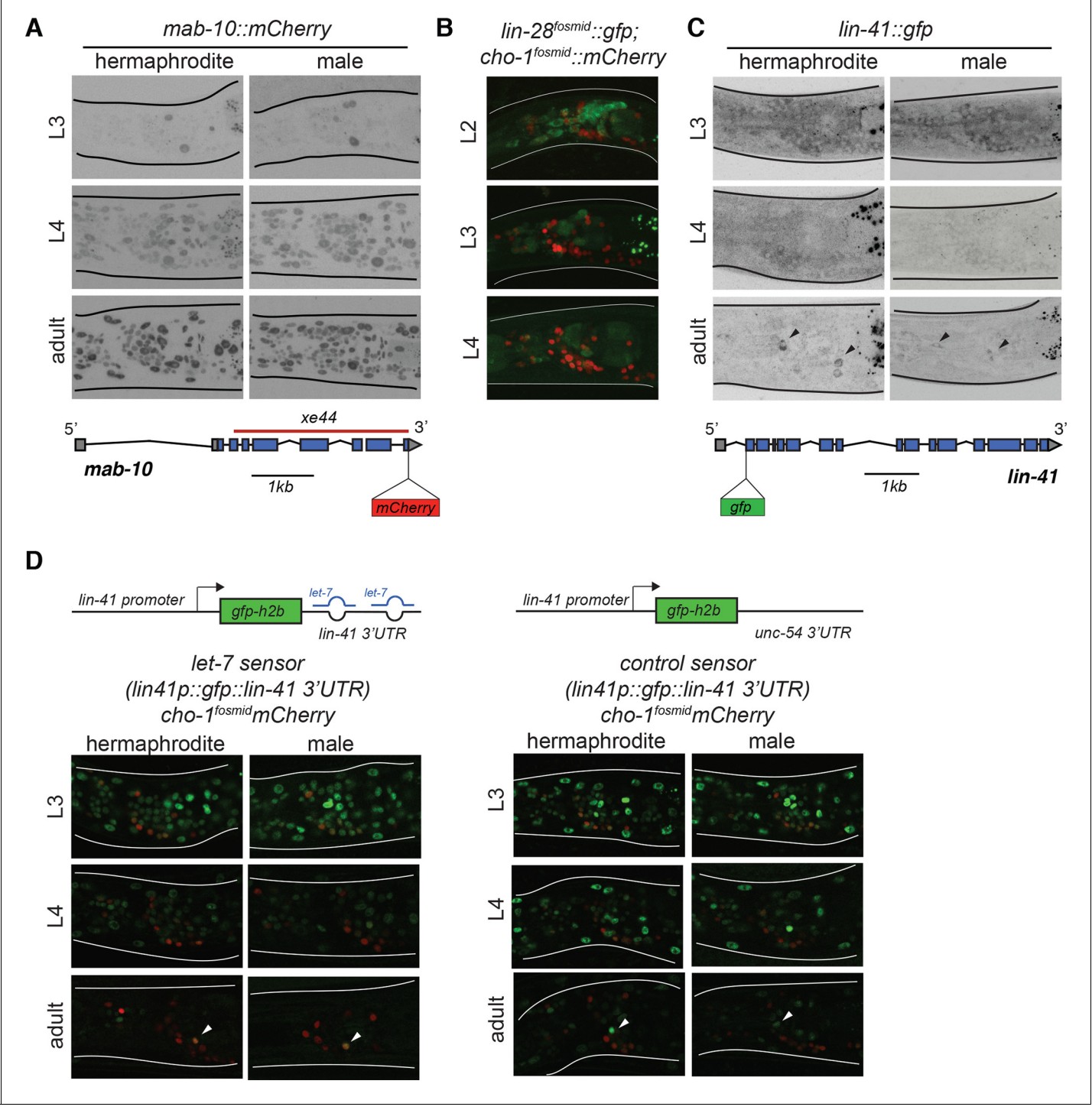

**Figure 2.** Temporal expression pattern of heterochronic pathway genes in the nervous system. (A) *mab-10::mCherry* expression pattern during larval development in both sexes. An endogenously tagged *mab-10* allele (*xe75*) showed ubiquitous expression starting in the L4 stage, including in the nervous system. Reporter expression was maintained in the adult. No sexually dimorphic differences were observed for *mab-10* expression. A cartoon for the *mab-10* locus is shown below the images. (B) *lin-28::gfp* expression pattern during early larval development. A *lin-28::gfp* fosmid-based reporter (*wgIs535*) showed ubiquitous expression at the L2 stage, including in the nervous system. Reporter expression was broadly downregulated at the L3 and L4 stages. The fosmid-based *cho-1/CHT* reporter (*otIs544*) was used to label cholinergic neurons. No sexually dimorphic differences were observed for *lin-28* expression. (C) *lin-41* expression pattern during larval development in both sexes. An endogenously tagged *lin-41* allele (*tn1541*) showed ubiquitous expression at the L3 stage, including in the nervous system. Reporter expression is broadly downregulated at the L4 stage. Expression of *lin-41::gfp* was maintained in two neuronal pairs in the head (black arrows). No sexually dimorphic differences were observed for *lin-41* expression. A

*Figure 2 continued on next page*

Figure 2 continued

cartoon for the *lin-41* locus is shown below the images. (D) *let-7* activity sensor and control sensor expression during larval development in both sexes. A *lin-41* promoter fusion driving *gfp* and fused to the *lin-41* 3'UTR (*xeSi182*), which is directly targeted by *let-7* miRNA, showed ubiquitous expression of *gfp* in the nervous system from the L1 to the L3 stage. At the L4 stage, reporter expression is broadly downregulated and only a few cells are labeled by GFP in the adult head, recapitulating the *lin-41::gfp* expression. The same *lin-41* promoter fusion driving *gfp* and fused to the *unc-54* 3' UTR (*xeSi202*) failed to be normally downregulated at the L4 stage showing that *lin-41* downregulation requires its 3'UTR being bound by miRNA *let-7*. Cholinergic neurons were labeled by a *cho-1/CHT* fosmid reporter (*otIs544*) driving *mCherry* allowing us to correlate the *lin-41* downregulation with AIM neurotransmitter switch in males at the L4 stage. No sexually dimorphic differences in *gfp* expression were observed for the *let-7* sensor and its control. Cartoons for each construct are shown above the images.

DOI: https://doi.org/10.7554/eLife.42078.006

UTR is not downregulated at the L4 and young adult stages (*Figure 2D*). We conclude that *let-7* activity is, as expected, temporally controlled in the nervous system. We did not observe any sexual dimorphisms of *lin-41* or *let-7* sensor expression in the nervous system.

To determine the expression pattern of the *lin-41* effector *lin-29* in the nervous system, we used strains in which the *lin-29* locus was tagged with *gfp*, using CRISPR/Cas9-mediated genome engineering. *lin-29* produces two distinct transcripts, *lin-29a* and *lin-29b* that differ at their 5'end; both contain five DNA-binding Zn finger domains (*Rougvie and Ambros, 1995*) (*Figure 3A*). We used a strain in which both isoforms were simultaneously tagged by introducing *gfp* at the 3' end of the locus (*Aeschimann et al., 2017*) (*Figure 3A*). In addition, we examined the expression pattern exclusively of the *b*-isoform, by introducing the same 3' terminal *gfp* tag in the *a*-isoform-specific mutant allele (*xe40*) via CRISPR/Cas9-mediate genome engineering (*Figure 3A*). To examine the expression pattern of the *a*-isoform, we inserted *gfp* at the 5' end of the *a*-isoform (*Figure 3A*). Generally, the expression pattern of the *gfp* allele that tags both isoforms matches the expression pattern reported using an antibody directed against both LIN-29 isoforms (*Figure 3B*) (*Bettinger et al., 1996*; *Euling et al., 1999*). As previously reported, we observed expression in head neurons and the ventral nerve cord. However, we found that this neuronal expression is almost completely restricted to the male and this male-specific expression can be entirely ascribed to the *a*-isoform (*Figure 3B–D*). The male-specific neuronal expression of *lin-29a* is not observed in male-specific neurons but is, interestingly, entirely restricted to sex-shared neurons (i.e. neurons that are generated in both sexes). Intriguingly, most of these neurons had not been previously reported to show sexually dimorphic gene expression patterns or functions. Specifically, 21 out of 116 sex-shared neuron classes express *lin-29a*, including seven sensory neuron classes in the head and tail (AWA, ASG, ASJ, ASK, ADF, PHB, PLM), seven interneuron classes (AVA, RIA, AIM, AVG, RIF, PVC, PVN), a few head and tail motor neurons (SAB, PDA, PDB) and most of the sex-shared ventral nerve cord motor neurons (dorsal and ventral A, B and D-type and AS neurons). Many of these LIN-29A expressing neurons are synaptically connected to one another (*White et al., 1986*) and some of the synaptic connections are known to be sexually dimorphic (*Jarrell et al., 2012*).

Expression of *lin-29a* in all these neurons in the male nervous system is precisely temporally controlled; it is first observed in the early L4 stage and persists throughout adulthood (*Figure 3D*). In addition to the male-specific neuronal expression of the *lin-29a* isoform, we also found that the *lin-29b* isoform is expressed in three postembryonically generated touch neurons (AVM, PVM, PDE), where its expression is not temporally regulated, that is, its expression is induced right after the neurons are born in the first two larval stages and expression is observed in both sexes (data not shown).

The late larval onset of expression of *lin-29* in non-neuronal cells has been reported to be controlled by the heterochronic pathway (*Bettinger et al., 1996*; *Moss et al., 1997*). To assess whether onset of male-specific, neuronal expression of the *lin-29a* isoform is controlled by *lin-41* and *let-7*, we analyzed *lin-29a* neuronal expression in these mutant backgrounds. Indeed, we found that in *lin-41(lf)* mutants *lin-29a::gfp* expression is observed prematurely including in the sex-shared neurons of the male (*Figure 4A*). As expected, in *let-7(lf)* and *lin-41(gf)* mutants (both of which maintain expression of LIN-41), *lin-29a::gfp* expression levels are significantly reduced compared to wild-type adult males (*Figure 4B*). This decrease in *lin-29a* expression is consistent with the similarity of loss-of-function phenotypes of *let-7* and *lin-29* null mutants and *lin-41* gain-of-function mutants in which the AIM neurotransmitter switch does not occur (*Figure 1*). Together with our previous biochemical analysis

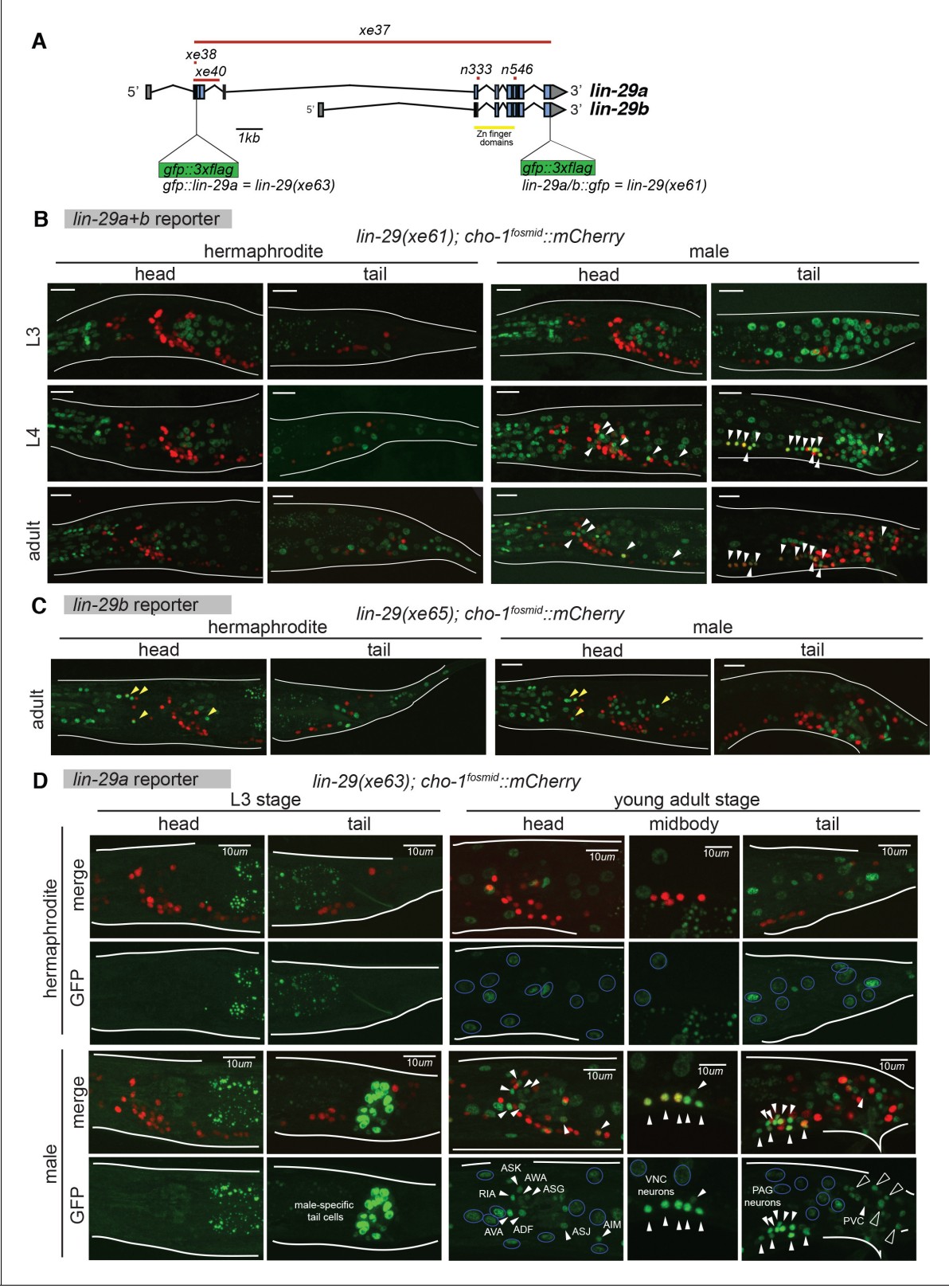

**Figure 3.** Temporal expression pattern of *lin-29* in the nervous system of both sexes. (**A**) Cartoon showing the *lin-29* locus. Alternative promoter usage generates two LIN-29 protein isoforms. Exons 1 – 4 are LIN-29A specific while exons 5 – 11, that include the Zn-finger DNA binding domain, are shared by both -A and -B isoforms. The *lin-29* locus was tagged using CRISPR/Cas9 genome engineering. *gfp* was inserted either at the C-terminal end (*lin-29a/b::gfp; xe61* allele) to tag both LIN-29A and B protein isoforms or at the N-terminal end (*gfp::lin-29a; xe63* allele) to tag only the LIN-29A isoform.
*Figure 3 continued on next page*

*Figure 3 continued*

Canonical alleles *n333* and *n546* as well as the newly generated alleles *xe37* (null), *xe38* (A-specific) and *xe40* (A-specific) are indicated. (B) *lin-29a/b* expression pattern during larval development in both sexes. Confocal images for endogenously tagged *lin-29a/b::gfp (xe61)* show that GFP is expressed in the pharynx at the L3 stage and onwards in both sexes (examination of young animals showed that pharyngeal expression starts at the L1 stage). Hypodermal GFP expression all along the body started at the end of the L3 stage in both sexes. At the L4 stage, we also detected GFP expression in neurons only in the male, many of which were also labeled by a cholinergic marker, a *cho-1/CHT mCherry* expressing fosmid (marked by arrows). Expression in these male neurons persisted in the adult stage. No neuronal expression was observed in the head nor tail in hermaphrodite animals. Scale bar: 10 µm. (C) *lin-29b* expression pattern at the young adult stage in both sexes. Confocal images for *gfp* tagged *lin-29b(xe40)* (this allele is given a new name, *xe65*, since it contains the *xe40* lesion plus the *gfp* insertion) show that GFP is expressed in head glia in both sexes (marked by yellow arrowheads). GFP is also observed in the pharynx and in tail cells in both sexes. Cholinergic *cho-1/CHT mCherry* expressing fosmid (*otIs544*) did not co-localize with GFP showing that LIN-29B is not expressed in neurons in the head nor tail in either sex (Note that LIN-29B is expressed in midbody neurons in both sexes not shown in this image). Scale bar: 10 µm. (D) *gfp::lin-29a* expression pattern during larval development in both sexes. Confocal images for endogenously tagged *gfp::lin-29a (xe63)* show that GFP is expressed in male-specific non-neuronal tail cells at the end of the L3 stage and onwards. No expression was observed in the head or tail of the hermaphrodite at this stage. No overlap between GFP and the cholinergic fosmid based reporter *cho-1/CHT (otIs544)* was observed at this stage. At the young adult stage, we observed GFP expression in neurons only in males (expression in male neurons started at the L4 stage), indicated by white arrows. No neuronal expression was observed in hermaphrodite neurons. Many GFP positive neurons in the male were also labeled by the *cho-1/CHT mCherry* expressing fosmid (*otIs544*). Cholinergic neurons expressing *lin-29a* included AIM, ASJ, AVA, ASK, AWA, ASG, RIA and ADF in the head, ventral nerve cord (VNC) motor neurons of the A, B, D and AS classes and the PVC interneuron in the tail. Neuronal GFP expression is indicated by arrows and neuronal identity is indicated for head and tail neurons in the GFP panels. GFP expression was observed throughout the hypodermis in both sexes, indicated with blue circles in the GFP panels. Hypodermis nuclei are larger. Cholinergic retrovesicular ganglion (RVG), ventral nerve cord (VNC) and pre-anal ganglion (PAG) neurons expressing *lin-29a* are shown in the head, midbody and tail respectively. Male-specific tail cells expressing *lin-29a* are marked with black arrows. Scale bar: 10 µm.

DOI: https://doi.org/10.7554/eLife.42078.007

(*Aeschimann et al., 2017*), we conclude that *let-7*-controlled LIN-41 directly represses *lin-29a* translation to control the precise timing of LIN-29A protein accumulation in the nervous system.

## The *lin-29a* isoform is required cell-autonomously and is sufficient to define the male-specific neurotransmitter switch in the AIM interneuron

To assess the functional relevance of the *lin-29a* isoform in controlling the AIM neurotransmitter switch, we generated two *lin-29a*-isoform specific alleles (*xe38* and *xe40*) using CRISPR/Cas9 genome engineering (*Figure 3A*) (*Aeschimann et al., 2018*). We indeed find that in both isoform-specific alleles the male-specific AIM neurotransmitter switch failed to occur (*Figure 5A*). A newly generated *lin-29* null allele that removes both isoforms (*Aeschimann et al., 2017*) also confirmed the results previously reported with the canonical null alleles tested (*Figure 5A*).

To assess whether *lin-29a* acts cell-autonomously in AIM to control its neurotransmitter switch, we expressed *lin-29a* under an AIM-specific promoter (*Serrano-Saiz et al., 2017a*). This driver is not only active in both sexes, but it is already active in embryos and early larval stages, hence allowing us to assess three key issues: (1) whether LIN-29A acts cell-autonomously, (2) whether LIN-29A can induce the neurotransmitter switch prematurely in males and (3) whether LIN-29A can ectopically induce the AIM neurotransmitter switch in hermaphrodites. We generated two independent transgenic lines that cell type-specifically express *lin-29a* in a *lin-29a (xe38 and xe40)* mutant background. We found that in adult male animals, the loss of the AIM neurotransmitter switch (as assayed by *cho-1* and *unc-17* induction and *eat-4* downregulation in AIM) is rescued by the transgene, thereby demonstrating cell autonomy of LIN-29A function (*Figure 5B*).

Examining juvenile males, before the onset of overt sexual maturation, we observed that the neurotransmitter switch is executed prematurely already by the first larval stage in these transgenic animals ectopically expressing LIN-29A in AIM neurons (*Figure 5B*). This sufficiency experiment indicates that the temporal control of *lin-29a* expression, and specifically the repression of *lin-29a* by the heterochronic pathway is required to prevent *lin-29a* from triggering the neurotransmitter switch precociously. Lastly, examining adult hermaphrodite animals expressing this construct, we found that *lin-29a* is sufficient to induce the neurotransmitter switch in the opposite sex (*Figure 5B*). We conclude that *lin-29a* is required and sufficient to control the AIM neurotransmitter switch.

To assess whether *lin-29a* function in AIM goes beyond controlling AIM's neurotransmitter switch, we examined another molecular marker of sexually dimorphic AIM differentiation, the orphan GPCR, *srj-54*, which is induced in the AIM neurons upon male sexual differentiation (*Portman, 2007*). We

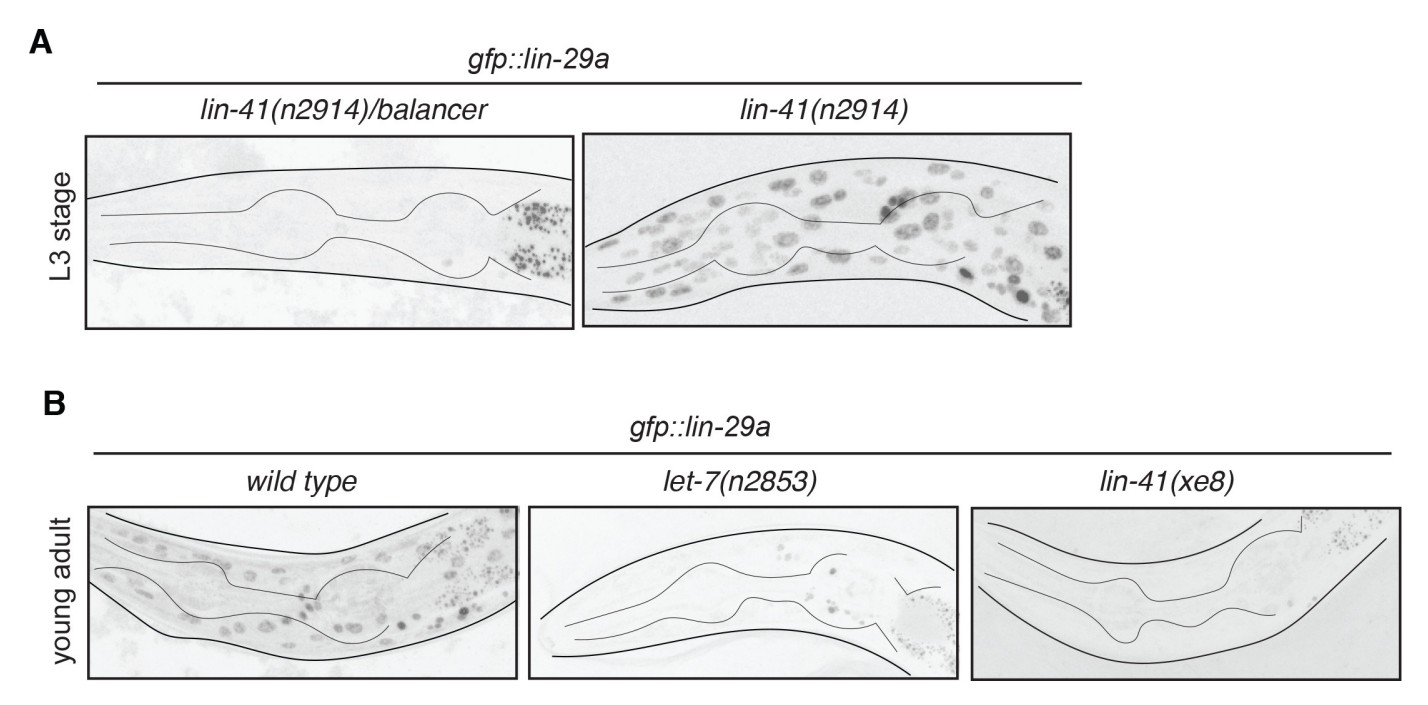

**Figure 4.** *lin-29a* neuronal expression is temporally regulated by *lin-41* and *let-7*. (**A**) *gfp::lin-29a* expression in the *lin-29a(xe63)* strain is precocious in the absence of *lin-41*. Expression was examined in synchronized L3 animals and compared between *lin-41(n2914)/balancer* control versus *lin-41(n2914)* loss-of-function males. Neuronal nuclei are more compact than the larger hypodermal nuclei. (**B**) *gfp::lin-29a* expression in the *lin-29a(xe63)* strain is downregulated in *let-7(lf)* and *lin-41(gf)* mutants. Expression was examined at the young adult stage in *let-7(n2853ts)* loss-of-function mutants and *lin-41 (xe8) gain-of-function* mutants, compared to control males. While control males showed expression of *gfp::lin-29a* in the hypodermis and nervous system (more compact nuclei), both mutants showed a severe downregulation of *gfp::lin-29a* expression in the hypodermis and neurons at the young adult stage. L1 *let-7(n2853ts)* animals were shifted to the restricted temperature (25°C) and imaged as adults, after 48hs.
DOI: https://doi.org/10.7554/eLife.42078.008

found that *srj-54* fails to be expressed in *lin-29a* mutants (**Figure 6A**) adding another AIM male-specific molecular feature that fails to be normally induced in the absence of *lin-29*. As we will describe further below, *lin-29a* mutants are also defective in an AIM-mediated sexual behavioral paradigm.

## *lin-29a* controls sexually dimorphic molecular features of other sex-shared neurons

The expression of *lin-29a* in a number of distinct neuron types indicated that *lin-29a* may play additional roles in controlling sexually dimorphic, that is, male-specific nervous system features. To this end, we assessed two other molecular markers that were previously found to be expressed in a sexually dimorphic manner in two distinct, sex-shared neuron classes, both of which express *lin-29a*. First, the *dmd-5* transcription factor was reported to be expressed in AVG neurons in adult males but not in hermaphrodites (**Oren-Suissa et al., 2016**). By analyzing the expression of a *dmd-5* reporter in *wild type* versus *lin-29a* mutants, we found that *lin-29a* is required for *dmd-5* expression in AVG in males (**Figure 6B**). Second, the *daf-7* gene, encoding a neuroendocrine TGFβ-like signal, is turned on during sexual maturation in the sex-shared ASJ neurons only in males (**Hilbert and Kim, 2017**). This induction is also abrogated in *lin-29a* mutants (**Figure 6B**). Non-sexually dimorphic expression of *dmd-5* and *daf-7* in other neuron types was not affected by *lin-29a*.

The electron micrographical reconstruction of the nervous system of *C. elegans* male revealed sexually dimorphic neuronal morphology and connectivity of several sex-shared neurons (**Jarrell et al., 2012**; **White et al., 1986**). Specifically, the *lin-29a*-expressing PDB motorneuron located in the pre-anal ganglion grows elaborate branches off from its cell body and initial segment of its posteriorly directed main process. This was only observed in the male but not in the hermaphrodite (**Jarrell et al., 2012**; **White et al., 1986**). These male-specific branches make extensive

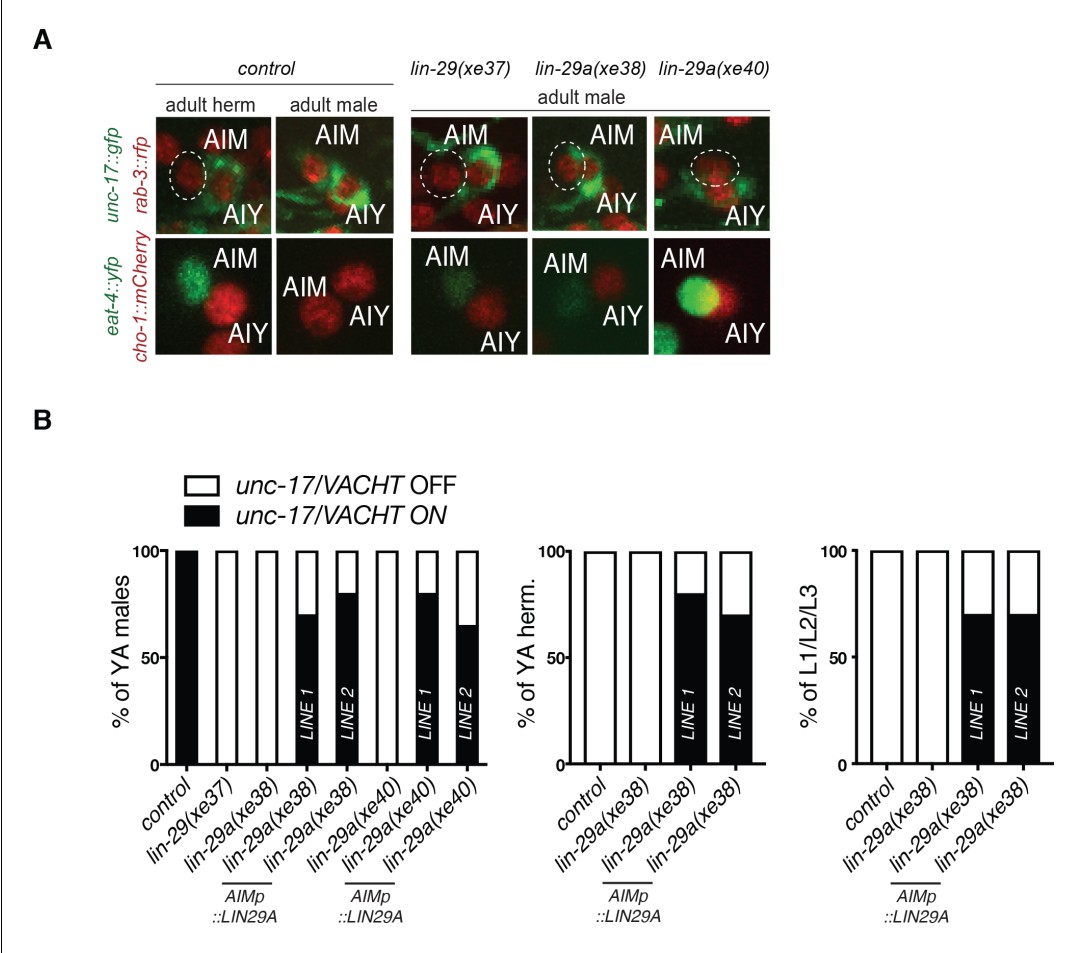

**Figure 5.** Cell-autonomy and sufficiency of *lin-29a* for the AIM neurotransmitter switch. (**A**) The neurotransmitter switch is blocked in three newly generated *lin-29* mutants. In adult control animals, the fosmid-based cholinergic reporters *unc-17/VACHT* (*otIs576*) and *cho-1/CHT* (*otIs544*) are expressed in AIM neurons of adult males but not hermaphrodites while the *eat-4/VGLUT* (*otIs518*) glutamatergic reporter is expressed in AIM neurons of adult hermaphrodites but not males. The AIM neurotransmitter switch is blocked in a newly generated *lin-29a/b(xe37)* null allele and in two *lin-29a*-specific mutants (*xe38* and *xe40*). In these *lin-29* mutant males, AIM fails to turn on cholinergic markers *unc-17/VACHT* (top panels) and *cho-1/CHT* (bottom panels) and expresses a pan neuronal maker *rab-3::rfp* (top panels) and the glutamatergic marker *eat-4/VGLUT* (bottom panels). A pan-neuronal *rab-3::rfp* (*otIs355*) reporter was used to label all neurons in the top panels. (**B**) AIM-specific LIN-29A expression is sufficient to cell-autonomously rescue the neurotransmitter switch in males and induce it in young larvae and the opposite sex. LIN-29A expression was driven under an AIM-specific promoter in *lin-29a* mutant animals: LIN-29A was expressed in *lin-29a(xe38)* (*otEx7316* and *otEx7317*) and in *lin-29a(xe40)* (*otEx7318* and *otEx7319*). Expression of cholinergic gene reporter *unc-17/VACHT* (*otIs576*) was examined to assay rescue and ectopic induction of the AIM neurotransmitter switch (n = 15).

DOI: https://doi.org/10.7554/eLife.42078.009

chemical and electrical synaptic contacts with both male-specific neurons, as well as sex-shared neurons (*Jarrell et al., 2012*). We corroborated the existence and reproducibility of male-specific branches by visualizing PDB with a reporter transgene (*Figure 6C*). We found that these branches grow specifically during sexual maturation in the L4 stage (*Figure 6C*), co-incident with the onset of *lin-29a* expression. Growth of these branches is disrupted in *lin-29a* mutants (*Figure 6C*). These findings demonstrate a role for *lin-29a* in regulating the temporal and sexually dimorphic acquisition of male-specific neuronal morphologies and predict that PDB fails to make its branch-specific synaptic contacts in the absence of *lin-29a*. We conclude that *lin-29a* affects multiple distinct types of sexual maturation events, from gene expression to anatomy, in distinct neuron classes.

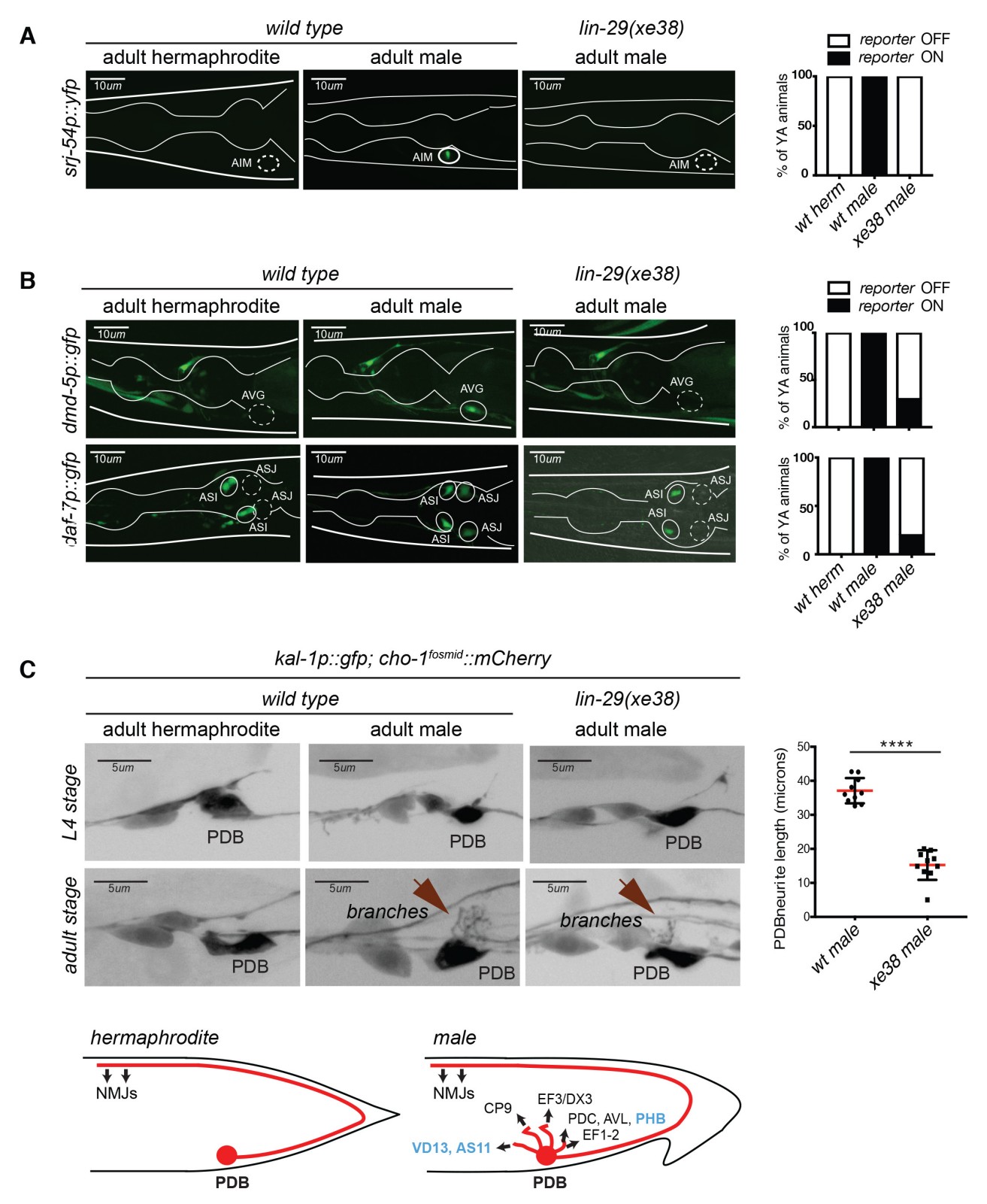

**Figure 6.** Male-specific molecular differentiation programs are lost in *lin-29a* mutants. (**A**) Male-specific expression of *srj-54* in AIM requires *lin-29a*. Expression of the GPCR *srj-54 (fsIs5)* promoter fusion is observed in adult males but not hermaphrodites in the AIM interneurons. Expression of *srj-54* was lost in all the *lin-29a(xe38)* mutant males. Quantification is shown on the right (n = 15). Scale bar: 10 μ*m*. (**B**) Male-specific expression of *dmd-5* in AVG requires *lin-29a*. Expression of the DM-containing transcription factor promoter fusion *dmd-5* (*pUL#JS9B3*) is observed in adult males but not

*Figure 6 continued on next page*

*Figure 6 continued*
hermaphrodites in the AVG interneuron. In the absence of *lin-29a*, expression is lost from AVG in 80% of the male worms (top panels); quantification is shown on the right (n = 15). The expression of the daf-7/TGFβ-like molecule promoter fusion is observed in adult males but not hermaphrodites and this expression is lost in the *lin-29a(xe38)* mutant males (bottom panels). Quantification is shown on the right (n = 15). Scale bar: 10 μm. (C) *lin-29a* is required for normal PDB branching. A *cis*-regulatory element from the *kal-1* locus is expressed in a number of neurons in the tail region in both sexes, including the PDB –inter and motorneuron (*Wenick and Hobert, 2004*). *kal-1::gfp* reporter expression showed that the PDB neuron process exhibits elaborate branching in the male after sexual maturation but not at earlier larval stages. In the absence of *lin-29a*, PDB branching is lost from adult males (bottom panels). Quantification of total neurite length is shown on the right (n = 10). PDB neurites establish extensive electrical and chemical synapses with sex-shared and male-specific neurons exclusively in adult males (schematized based on data from (*Jarrell et al., 2012*); synaptic partners of PDB present in both sexes are in blue and in black if sex-specific) . Scale bar: 5 μm.
DOI: https://doi.org/10.7554/eLife.42078.010

## *lin-29a* mutant males display defects in mating behavior

We next sought to broaden our understanding of *lin-29a* function in the nervous system by testing if *lin-29a* was required for the establishment of adult-specific sexually dimorphic behaviors. Critical for such an analysis is the fact that *lin-29a* mutant males do not display the obvious morphological and patterning defects displayed by the *lin-29* null allele. Specifically, the male tail, whose overall morphogenesis is disrupted in *lin-29* null mutants (*Euling et al., 1999*) appears morphologically normal in *lin-29a* mutants (*Figure 7A*). We precisely quantified other morphological differences between hermaphrodites and males using an automated visualization/measurement tool (*Yemini et al., 2013*) and found no overt morphological differences between wild-type and *lin-29a* mutant males (*Figure 7B*). The normal overall appearance of *lin-29a* mutant males allowed us to first ask whether *lin-29a* mutant males displayed mating-related behavioral defects. One aspect of mate attraction relates to the mating process itself, in which the male detects various sensory cues from the hermaphrodites to engage in direct contact (*Liu and Sternberg, 1995*). Previous work with the *lin-29* null mutant revealed multiple defects in these direct interactions and these defects were ascribed to spicule patterning defects (*Euling et al., 1999*). In spite of spicule patterning appearing normal in *lin-29a* isoform-specific mutants, we observed that these mutants were unable to mate (*Figure 7C*). We found that one aspect of mating behavior defective in these mutants is vulva location behavior (*Figure 7C*).

The AIM and ASJ neurons, both of which express *lin-29a* specifically in males, were previously implicated in mate-searching, which constitutes another aspect of mating behavior (*Barrios et al., 2012*; *Hilbert and Kim, 2017*). In this behavioral paradigm, well-fed single adult males leave a source of food to explore their environment in search for mating partners. We indeed found that *lin-29a* mutants are defective in mate searching behavior (*Figure 7D*). These defects are similar to the defects caused by loss of daf-7/TGFβ (*Hilbert and Kim, 2017*) and, consistent with this phenotypic similarity, we had described above that daf-7/TGFβ expression is lost in the ASJ neurons of *lin-29a* mutants. Nevertheless, we could partially rescue the mate searching defects of the *lin-29a* mutant males by restoring LIN-29A expression exclusively in the AIM neurons (*Figure 7D*), indicating that *lin-29a* may function in several distinct neurons to control mate searching behavior.

## Feminization of locomotor patterns of *lin-29a* mutant males

The male-specific expression of *lin-29a* in ventral nerve cord and head motor neurons and in a command interneuron pair made us consider whether *lin-29a* may be involved in controlling aspects of sexually dimorphic locomotor behavior. It has previously been shown that adult male and hermaphrodites display distinct locomotor behavior when crawling on a lawn of bacteria (*Mowrey et al., 2014*). We extended the analysis of sexually dimorphic adult-specific behavior using an automated worm tracker system which quantifies several hundred distinct postural and locomotory features of an animal (*Yemini et al., 2013*). We found a large number of sexually dimorphic features. While many of them were not affected in *lin-29a* mutant animals (*Figure 8* and *Supplementary file 1*), we found that *lin-29a* mutants displayed a striking feminization of a number of body posture and locomotory features. Postural differences included head bend, neck bend, midbody bend, hips bend, amplitude ratio and primary wavelength. (*Figure 8*). Other *lin-29a*-dependent locomotory features included midbody crawling amplitude and tail crawling amplitude (*Figure 8*). The lack of morphological differences between wild-type and *lin-29a* mutant males indicates that these locomotory

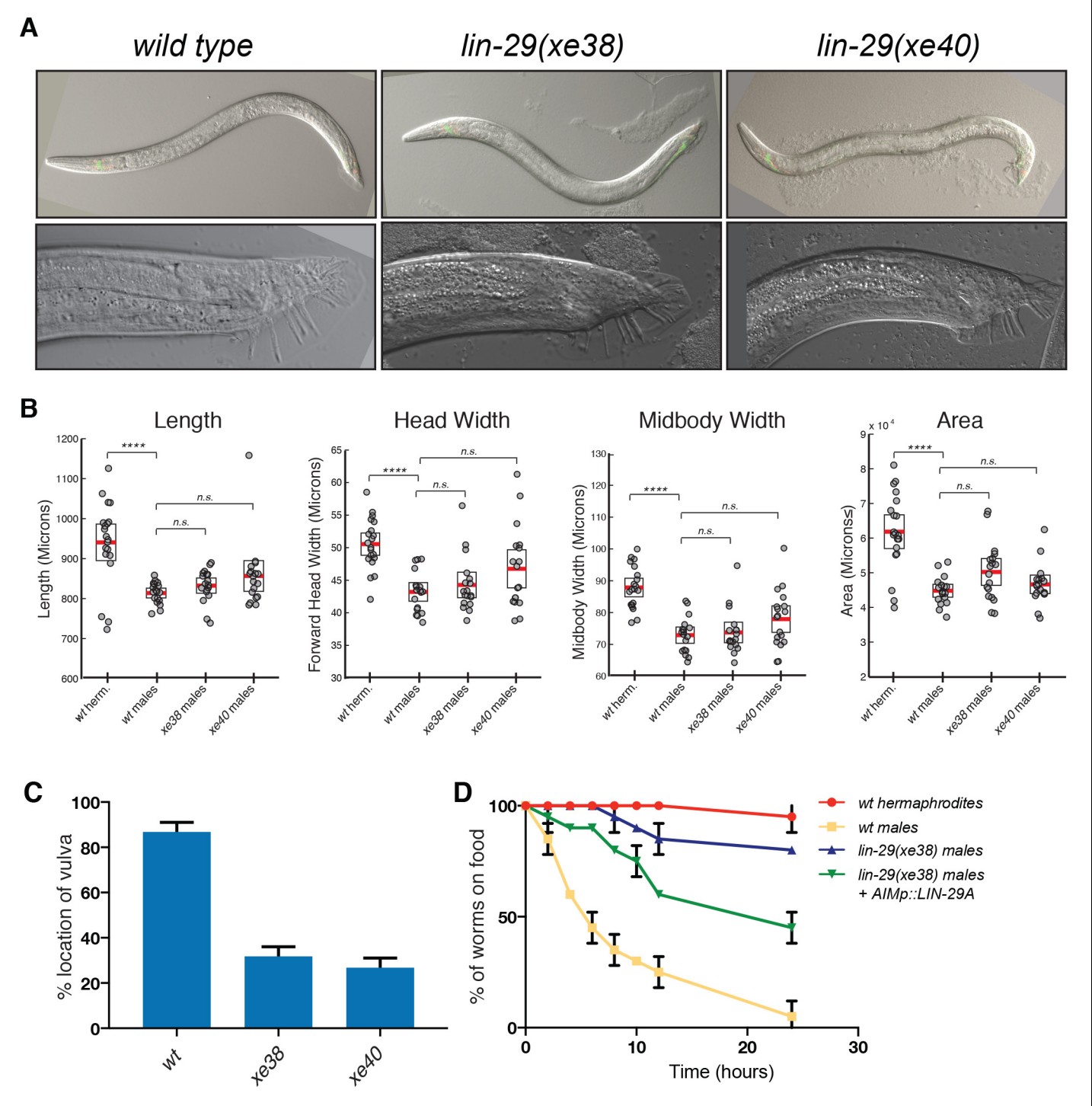

**Figure 7.** *lin-29a* is required for male mating behaviors. (A) *lin-29A* mutants show a normal male tail morphology. DIC images for young adult wild type and *lin-29a* mutants *xe38* and *xe40*. No morphological defects were detected in the young adult *lin-29a* mutant males compared to wild type. 20x images for young adult males are shown on the top panels. 63x images of the male tail are shown on the bottom panels. (B) WormTracker analysis of sexually dimorphic postural features showed no difference between wild-type and *lin-29a* mutant males. Hermaphrodites and males were tracked for 5 min and features describing body posture were analyzed using the WormTracker software. Adult hermaphrodites and males showed sex-specific differences in posture including length, head width, midbody width and area. When wild-type males were compared to *lin-29a xe38* and *xe40* mutant males no significant differences were found. (C) *lin-29a* is required for male mating. The ability of the of *lin-29a(xe38)* and *lin-29a(xe40)* mutant males to locate the hermaphrodite vulva is significantly reduced compared to *lin-29(+)* control males (n = 15). (D) *lin-29a* is required for male-specific mate searching behavior. Young adult males are followed over time and scored for movement beyond 3 cm away from the food source. *lin-29a(xe38)* mutant

*Figure 7 continued on next page*

*Figure 7 continued*
males failed to leave the food and search for mates and behaved similarly to wild-type hermaphrodites. *lin-29a* mate-searching defect could be partially rescued by restoring LIN-29A expression in AIM interneurons (*otEx7316*). Values plotted are an average of two independent experiments (n = 15 for each experiment).
DOI: https://doi.org/10.7554/eLife.42078.011

phenotypes are the result of neuronal defects rather than being a secondary consequence of overall body morphology. Taken together, these results show that *lin-29a* is required for the acquisition of male-specific postural and locomotory features and that the absence of *lin-29a* results in a feminization of these behavioral parameters.

We sought to determine in which of the neurons that express *lin-29a* in a sexually dimorphic manner *lin-29a* acts to control aspects of the locomotor behaviors described above. We particularly considered the so-called 'motor circuit', composed of a set of command interneurons and ventral nerve cord motor neurons (*Von Stetina et al., 2006*). Expression of *lin-29a* in the motor circuit, using a *cis*-regulatory element from the *unc-17/VAChT* locus rescued a defined subset of the feminized behavioral features (wavelength, crawling amplitude and midbody bend) (*Figure 8*). Other *lin-29a*-expressing neurons may be involved in controlling locomotory and postural features not rescued by this transgene.

## Intersectional control of the temporal, sexual and spatial specificity of *lin-29a* expression

Having established the function of *lin-29a* in sexually dimorphic development and function of the *C. elegans* nervous system, we sought to better understand the mode of *lin-29a* regulation. *lin-29a* shows a striking specificity of expression in three different dimensions. Within the nervous system, *lin-29a* is expressed in a temporally controlled manner (onset in embryonically born neurons only in the L4 stage), in a sex-specific manner (only in males) and it is expressed in a highly neuron type-specific manner. In as far as the first dimension, time, is concerned, we have described above that *lin-29a* expression in the nervous system is temporally controlled by the *lin-28/let-7/lin-41* heterochronic pathway. To assess the control of sex-specificity of *lin-29a* expression, we genetically removed *tra-1*, the master regulatory transcription factor of sex determination which is normally expressed in all hermaphroditic cells (XX sex chromosome genotype) where it suppresses male identities (*Schvarzstein and Spence, 2006*). We found that in *tra-1(e1488)* mutants, *lin-29a* expression becomes derepressed in XX animals (*Figure 9A*). Moreover, we induced the degradation of *tra-1* exclusively in the nervous system by force-expressing *fem-3*, which promotes TRA-1 protein destruction (*Starostina et al., 2007*), under the control of a panneuronal driver. We found that *lin-29a* expression is derepressed in the nervous system of the hermaphrodite, with the same pattern of cellular specificity seen normally in males (*Figure 9A*).

To address the cellular specificity of *lin-29a* expression, we considered the possibility that neuron class-specific terminal selector-type transcription factors that control the expression of neuron-class specific identity programs (*Hobert, 2016*) may also control the neuronal specificity of *lin-29a* expression. For example, the terminal differentiation program of the *lin-29a* expressing ASK neurons is controlled by the LIM homeobox gene *ttx-3/LHX2* and the identity of the AIM neuron is defined by the *unc-86/BRN3* POU homeobox gene (*Serrano-Saiz et al., 2013*). We found that *lin-29a* expression is eliminated in the ASK neurons of *ttx-3* mutants and in the AIM neurons of *unc-86* mutants (*Figure 9B*). The *unc-3*/COE-type transcription factor controls the identity of multiple neurons of the motor circuit, including the *lin-29a* expressing ventral nerve cord motorneurons, the SAB head motor neurons, the PDA and PDB tail motor neurons and the command interneurons AVA and PVC (*Kratsios et al., 2011*; *Pereira et al., 2015*). We found that in *unc-3* mutants *lin-29a* expression is lost in all these neurons (*Figure 9B*). We conclude that the complex *lin-29a* expression pattern can be explained by an intersectional mechanism. Temporal control is exerted via the heterochronic pathway, sexual specificity by the sex determination pathway and cellular specificity by terminal selector-type neuronal identity factors.

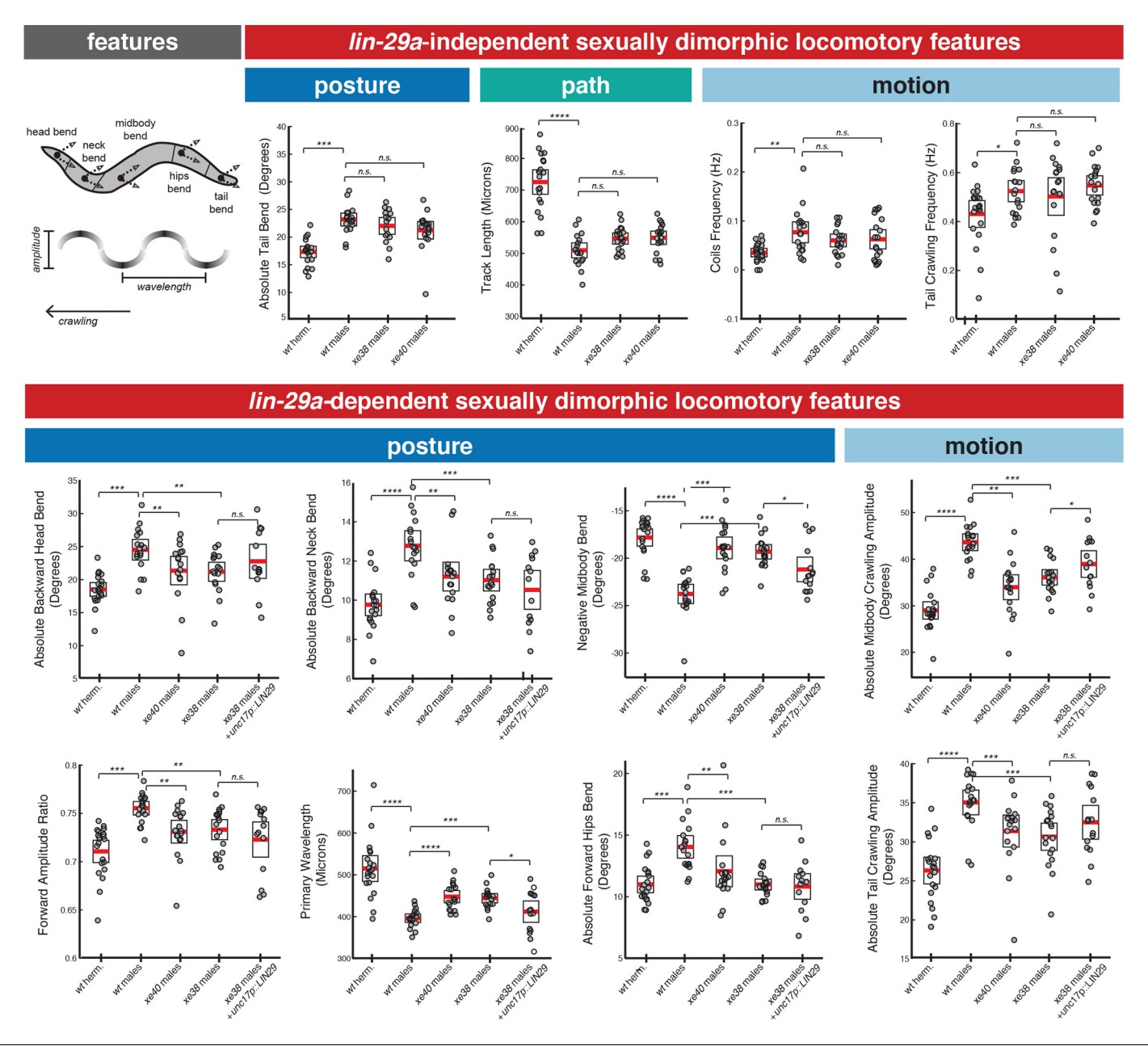

**Figure 8.** *lin-29a* is required for the establishment of male-specific features of locomotor behavior. There are *lin-29a* dependent and *lin-29a* independent adult-specific sexually dimorphic locomotory features. Cartoon of an adult *C. elegans* depicts the location of the body parts used for feature computation. The arrows for each body part represent the bend angle. Wavelength and amplitude while crawling are shown. Adult hermaphrodites and males showed sex-specific differences in posture, path and locomotory features that were not affected by *lin-29a* absence (top panel). A complete list of sexually dimorphic features not affected by *lin-29a* is provided in **Supplementary file 1**. In *lin-29a* mutant males, a number of sexually dimorphic features were feminized compared to wild-type males (bottom panel), including posture and locomotory features. Restoration of *lin-29a* expression in command interneuron AVA and ventral nerve cord motor neurons (*otEx7320*) partially rescued the feminization of two postural features (midbody bend and wavelength) as well as one locomotory feature (midbody crawling amplitude) (n = 20).
DOI: https://doi.org/10.7554/eLife.42078.012

## Other male-specific effectors of *lin-41* are also controlled in an intersectional manner

The sexually dimorphic expression pattern of *lin-29a* in the male nervous system does not extend to all sex-shared neurons that are known to display sexually dimorphic features. For example, the sex-

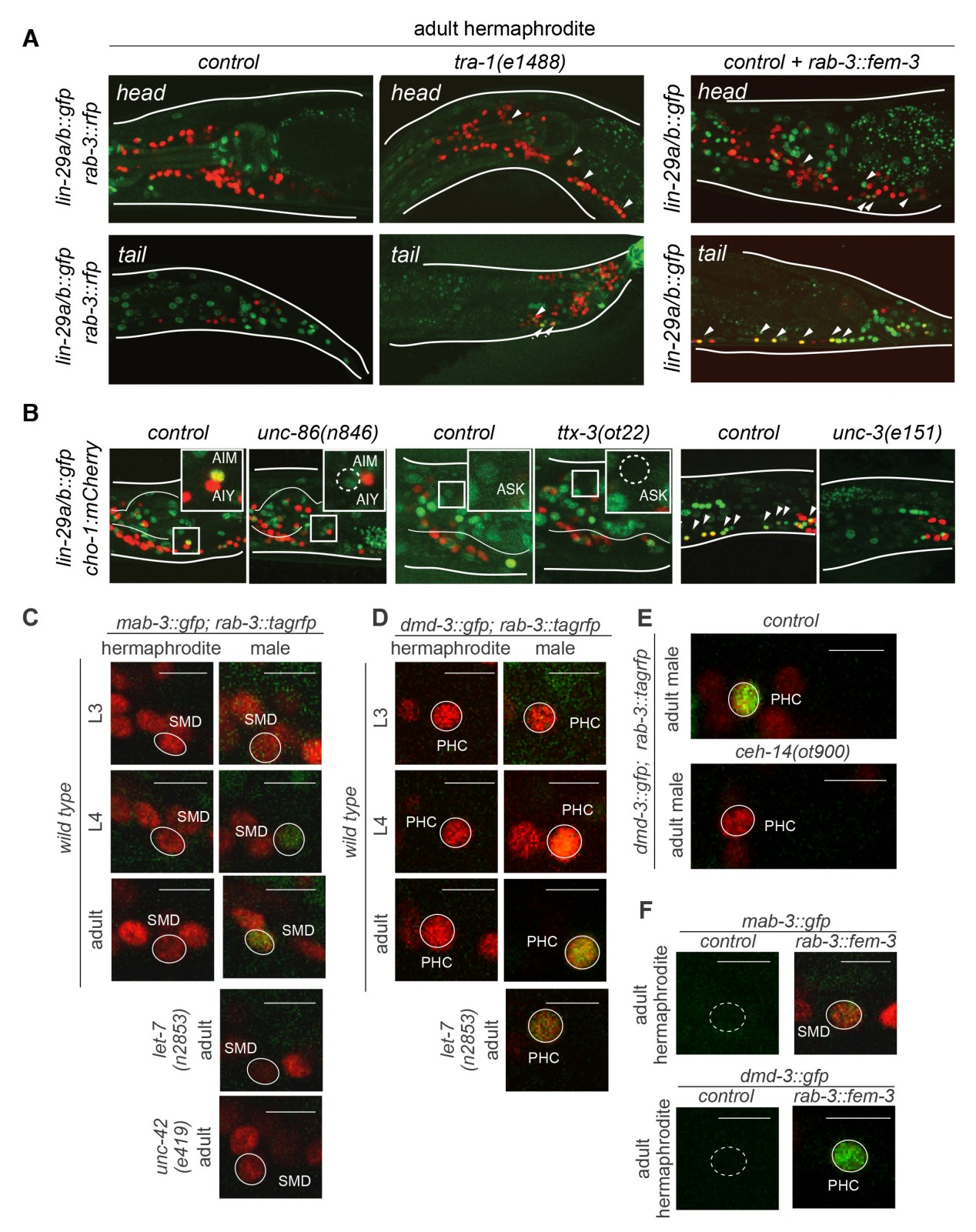

**Figure 9.** Intersectional control of distinct *lin-41* effectors via heterochronic, sexual (*tra-1*) and spatial (terminal selector) inputs. (**A**) LIN-29 neuronal expression is regulated by the sex-determination pathway. Mutant hermaphrodites for the downstream effector of the sex-determination pathway *tra-1* showed *lin-29a/b:gfp (xe61)* expression in hermaphrodite neurons in L4 and adult stages. Masculinization of the hermaphrodite nervous system by a pan-neuronal promoter driving *fem-3 (otEx7321)*, that represses *tra-1* function, induced *lin-29a/b::gfp (xe61)* expression in hermaphrodite neurons in L4

*Figure 9 continued on next page*

*Figure 9 continued*

and onwards. (**B**) *lin-29a/b::gfp* neuronal expression is lost in neuron-specific transcription factor mutants. The POU homeobox transcription factor *unc-86* is required for *lin-29a/b::gfp* expression in AIM interneurons. The LIM-HD transcription factor *ttx-3* is required for *lin-29a/b::gfp* expression in ASK. The COE/EBF transcription factor *unc-3* is required for *lin-29a/b::gfp* expression in AVA, SAB, DA, DB, VA, VB, PDA, PDB and PVC neurons. Cholinergic neurons are labeled with the *cho-1/CHT mCherry* fosmid reporter. (**C**) *mab-3::gfp* expression in the SMD neurons in control hermaphrodites, males and mutant animals. *mab-3::gfp* (*ot931*) expression was determined during larval development. GFP was first observed in the SMD neurons at the L4 stage and onwards, only in males. *mab-3::gfp* expression was lost from SMD neurons in a *let-7(n2853ts)* mutant male. L1 *let-7(n2853ts)* animals were shifted to the restricted temperature (25°C) and imaged as adults, after 48hs. *mab-3::gfp* expression was lost from SMD neurons in an *unc-42(e419)* mutant male. Scale bar: 5 μm. (**D**) *dmd-3::gfp* expression in the PHC neuron in control hermaphrodites, males and mutant animals. GFP was first observed in the PHC neurons at the L4 stage and onwards, only in males. *dmd-3::gfp* expression was decreased from PHC neurons in a *let-7(n2853)* mutant male. L1 *let-7(n2853)* animals were shifted to the restricted temperature (25°C) and imaged as adults, after 48hs. (**E**) *dmd-3::gfp* expression in PHC was lost in *ceh-14* null mutant males and in *let-7(n2853ts)* mutant males. L1 *let-7(n2853ts)* animals were shifted to the restricted temperature (25°C) and imaged as adults, after 48hs. (**F**) Masculinization of the hermaphrodite nervous system by a pan-neuronal promoter driving *fem-3* (*otEx7322 and otEx7323, respectively*), that promotes TRA-1 degradation, induced *mab-3::gfp* (*ot931*) expression in hermaphrodite SMD neurons and *dmd-3::gfp* (*ot932*) expression in hermaphrodite PHC neurons in L4 and onwards.

DOI: https://doi.org/10.7554/eLife.42078.013

shared PHC sensory neuron functionally and anatomically remodels during sexual maturation (*Serrano-Saiz et al., 2017a*), but does not express *lin-29a*. This suggests that *lin-29a* is possibly just one of several targets of LIN-41 in the context of male-specific nervous system differentiation. Previous whole animal RNA profiling of *let-7* and *lin-41* mutants, as well as RNA-binding assays demonstrated that LIN-41 also represses *mab-3* and *dmd-3* mRNA (*Aeschimann et al., 2017*). Both genes are members of the DMD family of sex-specifically expressed transcription factors (*Mason et al., 2008*; *Yi and Zarkower, 1999*) and were previously shown to be expressed in male-specific neurons and in a male-specific manner in distinct, but sex-shared neurons (*Mason et al., 2008*; *Serrano-Saiz et al., 2017a*; *Yi et al., 2000*). Their timing of expression had not been previously examined.

We *gfp*-tagged the endogenous *mab-3* and *dmd-3* loci using CRISPR/Cas9 genome engineering. Within the sex-shared nervous system, we found that *mab-3* is not only male-specifically expressed in the ADF neuron, as previously reported (*Yi et al., 2000*), but also in additional head neurons, including the embryonically generated SMD head motor neurons (*Figure 9C*). We found that like the male-specific, neuronal expression of *lin-29a*, male-specific neuronal expression of *mab-3* is precisely timed to commence at the L4 stage (*Figure 9C*). Similarly, the *gfp*-tagged *dmd-3* locus is expressed in a sex-shared neuron class, the PHC neurons (*Figure 9D*), where *dmd-3* acts to control the sexually dimorphic remodeling of PHC (*Mason et al., 2008*; *Serrano-Saiz et al., 2017a*). *dmd-3::gfp* expression in PHC also commences at the L4 stage (*Figure 9D*), at the time when PHC neurons remodel in a male-specific manner (*Serrano-Saiz et al., 2017a*). We corroborated our previous data of LIN-41 protein affecting *mab-3* and *dmd-3* expression (*Aeschimann et al., 2017*) by showing that the temporal specificity of nervous system expression of *mab-3* and *dmd-3* is indeed lost in *let-7* mutants (*Figure 9C–D*).

Like *lin-29a*, the male-specificity of *mab-3* and *dmd-3* expression in the nervous system is controlled by the *tra-1* transcription factor (*Figure 9E,G*) Moreover, the neuronal specificity of *mab-3* and *dmd-3* expression is, like *lin-29a*, also controlled by neuron-type specific terminal selectors. *unc-42* is a terminal selector transcription factor that controls terminal differentiation of the RMD and SMD neurons (*Pereira et al., 2015*), and we found that *mab-3* expression in these neurons fails to be induced in an *unc-42* mutant background (*Figure 9C*). Similarly, *ceh-14* is a terminal selector of PHC neuron identity (*Serrano-Saiz et al., 2017a*) and is required for *dmd-3* expression in PHC (*Figure 9F*). We conclude that *lin-41* operates through multiple distinct, male-specific effector genes and that the temporal, sexual and spatial specificity of expression of these distinct effectors is controlled by a similar, intersectional regulatory logic (*Figure 10A,B*).

## Discussion

We have described here that the timing of multiple distinct, neuron-type specific sexually dimorphic maturation events in the nervous system of *C. elegans* are controlled by a phylogenetically conserved regulatory cassette composed of ubiquitously expressed and temporally controlled genes, *lin-28* (a regulator of *let-7* miRNA processing), *let-7* (a miRNA) and its target *lin-41* (a translational

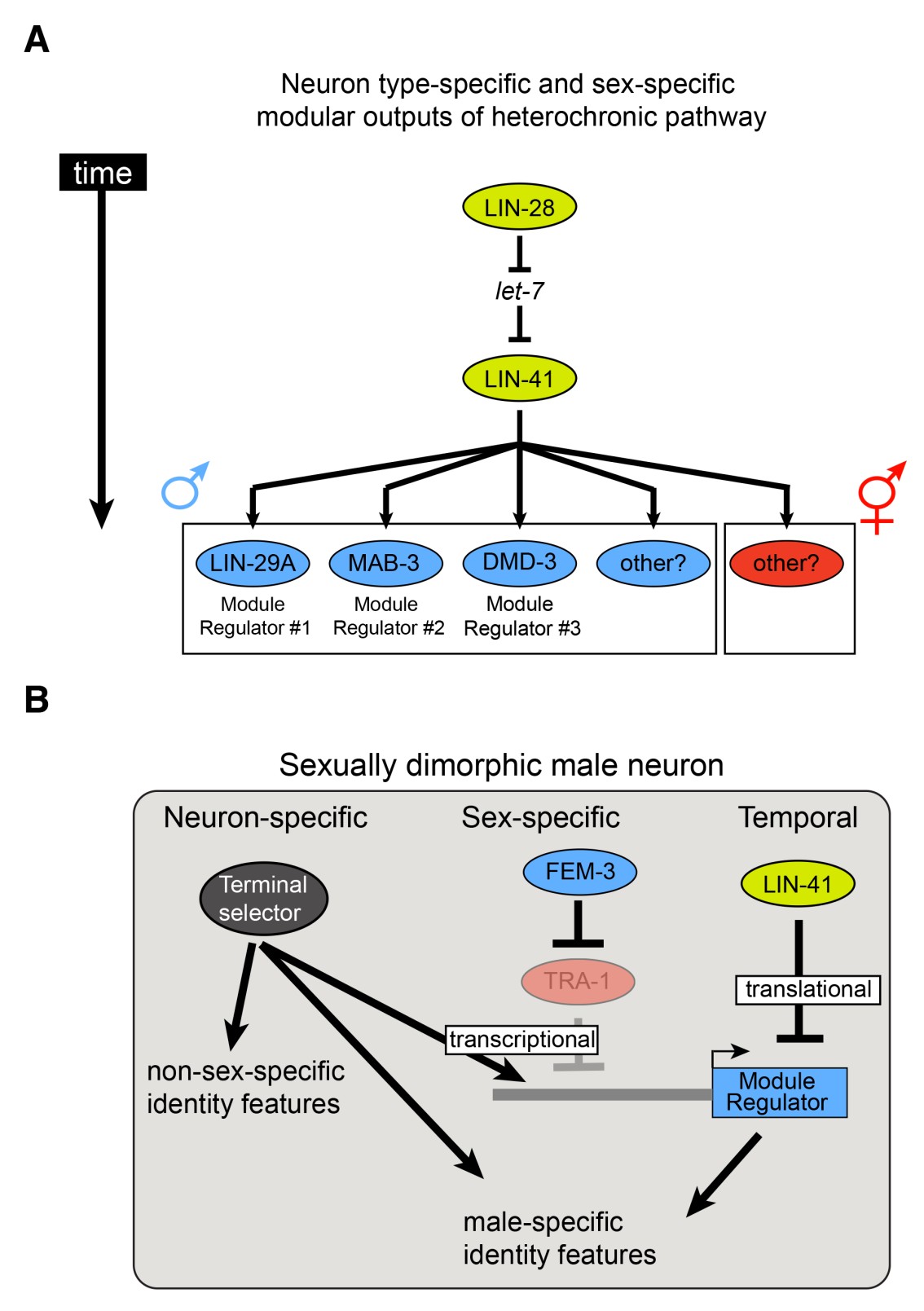

**Figure 10.** Timing mechanisms controlling sexually dimorphic nervous system differentiation during sexual maturation. (**A**) The evolutionarily conserved *lin-28/let-7/lin-41* heterochronic pathway controls the timing of the expression of multiple neuron-specific effectors in the male nervous system. The RNA-binding protein LIN-28 controls the expression of the miRNA *let-7*, which in turn controls expression of the RNA-binding protein LIN-41. LIN-41 *let-7*-mediated downregulation at the onset of sexual maturation de-represses LIN-41 targets LIN29A, MAB-3 and DMD-3. Expression of these LIN-41 *Figure 10 continued on next page*

*Figure 10 continued*

effectors in specific neurons in males, at the onset of sexual maturation, constitute effector modules directing the precisely timed nervous system masculinization. We hypothesize that similar LIN-41 effectors, not identified yet, could be acting in other neurons in the male nervous system and in the hermaphrodite nervous system. (B) Temporal, sex-specific and neuron-specific intersectional control of nervous system differentiation during sexual maturation. A hypothetical male neuron is shown. As discussed above, LIN-41 repression of the heterochronic pathway is relieved at the onset of sexual maturation, allowing neuron-specific effector proteins to be expressed. In males, degradation of the ubiquitously expressed transcriptional repressor TRA-1 allows expression of male-specific regulatory effectors (blue boxes same as panel A). The neuron type-specificity of regulatory effector gene expression is determined by terminal selector transcription factors that control the expression of many other neuronal identity features. Hence, terminal selectors define a neuron type-specific, but permissive state whose sex- and time-specificity is controlled by factors that antagonize the effect of terminal selectors on regulatory effector genes (transcriptionally – TRA-1; translationally – LIN-41). Note that features that are controlled by sex/time-specific regulatory effectors, exemplified by LIN-29A regulation of neurotransmitter pathway genes, are also controlled by terminal selectors, thereby constituting a 'feedforward' regulatory architecture.

DOI: https://doi.org/10.7554/eLife.42078.014

regulator). The conservation of the deployment of this pathway in the timing of sexual maturation is remarkable. Genetic manipulations in mice, as well as genome-wide association studies of humans with abnormally timed puberty demonstrate that *lin-28* and *let-7* also control timing of sexual maturation (puberty) onset in mammals (*Chen et al., 2017*; *Corre et al., 2016*; *He et al., 2009*; *Ong et al., 2009*; *Perry et al., 2009*; *Sulem et al., 2009*; *Zhu et al., 2010*). This conclusion was further corroborated by the finding that mutations in human makorin, the orthologue of *lep-2*, a regulator of *lin-28* activity in *C. elegans* (*Herrera et al., 2016*), also affect the timing of puberty in humans (*Abreu et al., 2013*; *Bulcao Macedo et al., 2014*). Similarly, in the fly nervous system, *lin-28* and *let-7* are involved in controlling the timing of pupal to adult transitions (*González-Itier et al., 2018*; *Sempere et al., 2003*; *Wu et al., 2012*) that are accompanied by sexual maturation in the nervous system, as assessed by expression of the male-specific isoform of Fruitless, a master regulator of male-specific nervous system differentiation (*Lee et al., 2000*). However, a regulatory linkage between the heterochronic pathway and transcriptional regulators of sexual differentiation (hormone receptors in vertebrates or Fruitless in flies) has not yet been established in these systems.

Apart from suggesting a remarkable conservation of timing mechanisms of sexual maturation, we have provided insights into the mechanisms by which the *lin-28/let-7* axis affects the timing of sexual maturation. The *let-7* miRNA is known to have a plethora of distinct mRNA targets (*Büssing et al., 2008*), but the one that we define here as being relevant in the context of sexual maturation is the translational regulator *lin-41* (*Slack et al., 2000*). *C. elegans lin-41* had previously been implicated as an output of *let-7* in the timing of skin cell maturation events (*Slack et al., 2000*) as well as vulval patterning (*Ecsedi et al., 2015*). *lin-41* is also highly conserved in vertebrates where its expression appears to be temporally regulated during embryogenesis with its expression being complementary to *let-7*, as observed in *C. elegans* (*Schulman et al., 2005*). The dynamics of *lin-28*, *let-7* and *lin-41* expression has not yet been comprehensively examined in the context of sexual maturation in the vertebrate brain, but notably, the expression of Lin28 and Let7 in the hypothalamus, a site of release of peptides that control pubertal onset, appears to be temporally controlled (*Sangiao-Alvarellos et al., 2013*). In *Drosophila*, *let-7* expression has also been found to be temporally regulated; its expression increases during late pupal stages (*Faunes and Larraín, 2016*). This is the time at which at least some parts of the *Drosophila* nervous system sexually mature (*Lee et al., 2000*), but a role of *Drosophila let-7* in the timing of sexual maturation has not yet been assessed.

One key remaining question, raised by our studies in *C. elegans*, is how *lin-28/*Lin28 function is precisely timed in the nervous system. Surprisingly, we found that a previously described upstream regulator of *lin-28*, the *lin-4* miRNA (*Moss et al., 1997*), does not affect male-specific sexual maturation in the nervous system, suggesting the existence of alternative pathways that determine when *lin-28*'s function in controlling *let-7* processing is downregulated. Recent work in *C. elegans* has discovered LIN-28 protein degradation, mediated by the conserved LEP-2/Makorin protein, as another means to control *lin-28* function (*Herrera et al., 2016*). It remains to be explored how LEP-2/Makorin-mediated LIN-28 degradation is temporally controlled.

We showed here that the key target of *let-7*, the *lin-41* translational regulator, relays temporal information in a modular manner to distinct, neuron class-specific outputs (*Figure 10A*). Among them is the phylogenetically conserved LIN-29 Zn finger transcription factor (whose vertebrate

orthologs ZNF362 and ZNF384 are completely unstudied), which we identified as being sexually dimorphically expressed in one fifth of all sex-shared neuron classes. Two other targets of *lin-41* that are relevant for sexual maturation are two male-specifically expressed DM domain transcription factors (*Aeschimann et al., 2017*) (this paper). Neurons that express these sexually dimorphic output regulators (*lin-29, mab-3, dmd*-3) act at distinct levels of information processing, from sensory, to inter- to motorneurons. Intriguingly, many of the sex-shared, LIN-29(+) interneurons are synaptically connected and some of these connections are sexually dimorphic. Some of the neurons that express LIN-29 or MAB-3 in a male-specific manner were not previously shown to display any sexual dimorphisms, demonstrating that the *C. elegans* nervous system may be even more sexually dimorphic than previously appreciated.

*lin-29, mab-3* and *dmd-3* expression mark many, but not quite all sex-shared neurons with sexually dimorphic features. We indeed found evidence for two other *dmd* genes whose expression is both sexually and temporally controlled in yet distinct neuron classes (*Oren-Suissa et al., 2016*) (unpubl. data). Altogether, these findings suggest a scenario in which a ubiquitously expressed temporal coordinate system (heterochronic pathway), intersects with a ubiquitously expressed sexual coordinate system (sex determination pathway) to control the sexual and temporal specificity of a number of modular, neuron-type specific output systems that control sexual dimorphisms in distinct cellular circuit modules (*Figure 10A,B*). This modular organization displays striking conceptual similarities with the modular organization of sexual dimorphic control mechanisms in vertebrates (*Yang and Shah, 2014*).

How is the neuron type-specificity of the modular outputs (*lin-29, mab-3, dmd-3* etc.) of the temporal and sex-determination system controlled? We provided evidence here that the neuron-type specificity of the sex-specific regulator is controlled by neuron-type specific terminal selector transcription factors which define the overall differentiated state of individual neuron classes (*Hobert, 2016*) (*Figure 10B*). For example, *unc-3* is required for many (non-sexually dimorphic) aspects of motor neuron differentiation in the ventral nerve cord and we find that *unc-3* also is required to target *lin-29* expression specifically in ventral nerve cord neurons. Like the regulation of *lin-29* by the temporal cue *lin-41*, the neuron-type specific regulation of *lin-29* could potentially be direct since we noted several phylogenetically conserved binding sites for UNC-3 protein in the *lin-29* locus. The regulation of sex-specific output regulators, like *lin-29*, by terminal selectors reveals the potential existence of a specific gene regulatory network architecture. For example, in the male AIM neurons, *unc-86* and *ceh-14* are required for the induction of *unc-17* and *cho-1* expression (*Pereira et al., 2015*). Both are apparently not sufficient to induce *unc-17* and *cho-1* expression in hermaphrodites, but via the *unc-86*-mediated induction of *lin-29* expression in the AIM neurons in males, *unc-86* and *ceh-14* are now apparently able to induce, with the help of *lin-29*, the expression of *unc-17* and *cho-1*. Further biochemical studies will be required to corroborate the existence of such feedforward architecture (*Figure 10B*). In conclusion, we have identified an intersectional regulatory mechanism in which temporal cues, sexual cues and neuron-type specific cues define the onset of expression of sex-specific, neuron-type specific 'output regulators' which define sexual dimorphic features of the *C. elegans* nervous system (*Figure 10*).

## Materials and methods

### *C. elegans* strains

Worm strains used in this study are listed in *Table 1*. The wild-type strain was Bristol N2. Worms were grown on nematode growth media (NGM) seeded with bacteria (OP50) as a food source. To synchronize worms 10 adult gravid hermaphrodites were transferred to a new plate with OP50. Worms were allowed to lay eggs for approximately two hours and adult gravid hermaphrodites were removed afterwards.

*srj-54::yfp* is a PCR fragment that was constructed using standard PCR fusion methods by fusing fragment A, comprising 664 bp upstream of the predicted ATG of *srj-54* and the first four codons of *srj-54* coding sequence, to fragment B, comprising *yfp* fused to the *unc-54* 3′ UTR. *srj-54::yfp* was injected with *unc-122::gfp* into *him-5(e1490)* to generate a stable extrachromosomal array. This array was integrated into the genome using standard UV irradiation methods followed by several rounds of backcrossing to *him-5(e1490)* to generate *fsIs5*.

**Table 1.** Strain list.

| Strain name | Genotype | Relevant DNA on array or single copy transgene |
|---|---|---|
| DR466 | *him-5(e1490)* | |
| MT1524 | *lin-28(n719)* | |
| MT7897 | *lin-41(n2914)/unc-29(e1072); lin-11(n1281)* | |
| MT7626 | *let-7(n2853)* | |
| HW1814 | *lin-41(xe8/bch28); him-5(e1490)* [1] | |
| VT132 | *lin-29a/b(n333)/mnC1; sqt-1(sc13)* | |
| CB4037 | *glp-1(e2141)* | |
| HW1672 | *lin-29a/b(xe37)* | |
| HW1693 | *lin-29a(xe38)* | |
| HW1695 | *lin-29a(xe40)* | |
| HW1698 | *mab-10(xe44)* | |
| CB2823 | *tra-1(e1488)/eDp6* | |
| MT1859 | *unc-86(n846)* | |
| OH161 | *ttx-3(ot22)* | |
| CB151 | *unc-3(e151)* | |
| CB419 | *unc-42(e419)* | |
| OH15422 | *ceh-14(ot900)* [2] | |
| OH15568 | *unc-17(ot907[unc-17::mKate2::3xflag])* | |
| DG3913 | *lin-41(tn1541[lin-41::gfp])* | |
| HW1822 | *lin-29(xe61[lin-29a/b::gfp::3xflag])* | |
| HW2224 | *lin-29(xe63 [gfp::3xflag::lin-29a]); him-5(e1490)* | |
| HW1835 | *lin-29(xe65 [lin-29b::gfp::3xflag; lin-29(xe40)])* | |
| HW2047 | *mab-10(xe75[mab-10::flag::mCherry])* | |
| HW2225 | *lin-29(xe63); him-5(e1490); let-7(n2853)* | |
| HW2342 | *lin-29(xe63); him-5(e1490); lin-41(n2914)/unc-29(e1072); lin-11(n1281)* | |
| HW2344 | *lin-29(xe63); him-5(e1490); lin-41(xe8/bch28)* | |
| OH15732 | *mab-3(ot931[mab-3::3xflag::gfp])* | |
| OH15733 | *dmd-3(ot932[dmd-3::3xflag::gfp])* | |
| OH10689 | *otIs355* | *rab-3::nls::tag_rfp* |
| OH12543 | *otIs534* | *cho-1 fosmid::sl2::yfp::h2b* |
| OH12655 | *otIs544* | *cho-1 fosmid::sl2::mCherry::h2b* |
| OH13083 | *otIs576* | *unc-17 fosmid::gfp* |
| OH11124 | *otIs388* | *eat-4 fosmid::sl2::yfp::h2b* |
| OH12496 | *otIs518* | *eat-4 fosmid::sl2::mCherry::h2b* |
| OP535 | *wtIs535* | *lin-28 fosmid::gfp* |
| HW2172 | *xeSi182* | *plin-41::gfp(pest)::h2b::lin-41 3'UTR* |
| HW2173 | *xeSi202* | *plin-41::gfp(pest)::h2b::unc-54 3'UTR* |
| UR219 | *fsIs5* | *srj-54p::yfp* |
| UL2497 | *Ex(dmd-5p::gfp)* | *dmd-5p::gfp (pUL#JS9B3 plasmid)* |
| FK181 | *ksIs2* | *daf-7p::gfp* |
| EB2509 | *dzIs75 II* | *kal-1p::gfp* [3] |
| OH15741 | *lin-29(xe38) + otEx7316* | *eat-4p::lin-29a* |
| OH15742 | *lin-29(xe38) + otEx7317* | *eat-4p::lin-29a* |
| OH15743 | *lin-29(xe40) + otEx7318* | *eat-4p::lin-29a* |

*Table 1 continued on next page*

*Table 1 continued*

| Strain name | Genotype | Relevant DNA on array or single copy transgene |
| --- | --- | --- |
| OH15744 | lin-29(xe40) + otEx7319 | eat-4p::lin-29a |
| OH15745 | lin-29(xe38) + otEx7320 | unc-17p::lin-29a |
| OH15746 | lin-29(xe61) + otEx7321 | rab-3p::fem-3::mCherry |
| OH15748 | mab-3(ot931) + otEx7322 | rab-3p::fem-3::mCherry |
| OH15749 | dmd-3(ot921) + otEx7323 | rab-3p::fem-3::mCherry |

[1]The *xe8* gain of function and *bch28* null alleles have been described in **Ecsedi et al. (2015)** and in **Katic et al. (2015)**. These alleles were maintained in *trans* to improve health of the animals. Offspring homozygous for either allele were scored for phenotypic traits

[2]**Bayer and Hobert (2018)**.

[3]The *kal-1 gfp* promoter fusion corresponds to 'promoter G' from **Wenick and Hobert, 2004**

DOI: https://doi.org/10.7554/eLife.42078.015

For neuron-specific expression of *lin-29a*, promoter constructs driving *lin-29a* cDNA were generated by Gibson assembly or RF cloning. cDNA was amplified from pDH06, kindly provided by Dr. Horvitz (**Harris and Horvitz, 2011**). A 1 kb of the *unc-17/VACHT* promoter that drives expression in AVA, SAB, DA, DB, VA, VB, PDA and PDB neurons (our unpublished data) was amplified by PCR from the fosmid construct (FW AATGAAATAAGCTTGCATGGTATACACCAATCATTTCTCC and REV TCCTCTAGAGTCGACCTGCAGATAATTTAATTAAAATTGAGTTCCAAC). The PCR fragments were assembled into pPD95.75 by Gibson. For AIM-specific expression a fragment of the *eat-4* locus was used to drive LIN-29A expression. The *lin-29a* cDNA was inserted to replace the *fem-3* cDNA in this construct (**Pereira et al., 2015**). For nervous system masculinization, a *rab-3* pan-neuronal promoter driving *fem-3::mCherry* was used (**White et al., 2007**). Transgenic lines carrying extrachromosomal arrays were generated by germline injection with 20 ng/ul of plasmid DNA and *unc-122::gfp* as a co-injection marker, also at 20 ng/ul.

## Genome engineering

*Isoform-specific gfp::3xflag tagging of endogenous lin-29 using CRISPR/Cas9.* In order to obtain the *lin-29(xe65)* allele, the following mix was injected into *lin-29a(xe40)* mutant worms: 50 ng/μl pIK155, 100 ng/μl of pIK198 with a cloned sgRNA (atattatttatcagtgattg), 2.5 ng/μl pCFJ90, 5 ng/μl pCFJ104 and 10 ng/μl pFA26 (pDD282 with cloned homology arms). Recombinants were isolated according to the protocol by Dickinson *et al.* (**Dickinson et al., 2015**), verified by DNA sequencing and outcrossed three times. The plasmid for homologous recombination (pFA26) was previously described (**Aeschimann et al., 2017**). The *lin-29(xe63)* allele specifically tags *lin-29a* with GFP::3xFLAG at the N-terminus (**Aeschimann et al., 2018**).

*Tagging of endogenous mab-10 with flag::mCherry using CRISPR/Cas9.* In order to tag *mab-10* with *flag::mCherry* at the C-terminus, the following mix was injected into *unc-119(ed3)* worms (**Dickinson et al., 2015**; **Katic et al., 2015**): 25 ng/ul pIK155, 65 ng/ul of pIK198 with a cloned sgRNA (gctcccggaatcttgaagct), 10 ng/ul pIK127 (Peft-3::gfp::h2b), 5 ng/ul Pmyo-3::gfp and 65 ng/ul pIK284. The plasmid serving as a template for homologous recombination (pIK284) was obtained using the SapTrap protocol (Schwartz et al., 2016), by combining SapTrap component vectors pMLS257, pMLS291 and pMLS382 with two pairs of annealed oligos (oIK1052/oIK1053, oIK1054/oIK1055). Wild-type moving recombinants were isolated, verified by DNA sequencing and outcrossed three times. The *mab-10::mCherry* allele is not fully functional since *mab-10(xe75)* and *mab-10(xe44)* both enhance the bursting phenotype of *let-7(n2853ts)* (data not shown).

*lin-29, lin-29a and mab-10 null mutant alleles.* In order to specifically mutate *lin-29a* without affecting expression of *lin-29b*, we introduced deletions into the coding exons specific to *lin-29a*, at the same time introducing a frame-shift in the downstream *lin-29a* reading frame. The *lin-29a(xe38)* allele is a 10 bp deletion in exon 2, resulting in a frame-shift and a stop codon in exon 3. It was obtained by injecting a single sgRNA (gctggaaccaccactggctc) and has the following flanking sequences: 5' gaatagctggaaccaccact – *xe38* deletion – ctaccacccatttggtgtt 3'. The *lin-29a(xe40)* allele is a 1102 bp deletion covering exons 2–4, introducing a frame-shift in the *lin-29a* reading frame with a predicted stop codon in exon 6 (**Aeschimann et al., 2018**). We used deletions of almost the entire

coding regions as null alleles for *lin-29* and *mab-10*: The *lin-29(xe37)* allele is a 14.8 kbp deletion spanning *lin-29* exons 2–11 and the *mab-10(xe44)* allele is a 2.9 kbp deletion spanning *mab-10* exons 3–9 (*Aeschimann et al., 2018*).

To obtain mutant worm lines, wild-type worms were injected with a mix containing 50 ng/µl pIK155, 100 ng/µl of each pIK198 with a cloned sgRNA, 5 ng/µl pCFJ90 and 5 ng/µl pCFJ104, as previously described (*Katic et al., 2015*). Single F1 progeny of injected wild-type worms were picked to individual plates and the F2 progeny screened for deletions using PCR assays. After analysis by DNA sequencing, the alleles were outcrossed three times to the wild-type strain.

*Tagging of endogenous unc-17 with 3xflag::mKate2 using CRISPR/Cas9.* The *unc-17* locus was tagged at the C-terminus as previously described (*Dickinson et al., 2015*; *Katic et al., 2015*).The plasmid serving as a template for homologous recombination was cloned using Gibson and included 700 bp-long homology arms. Wild-type moving recombinants were isolated, verified by DNA sequencing and outcrossed three times.

*Tagging of endogenous mab-3 and dmd-3 with 3xflag::gfp using CRISPR/Cas9.* The *mab-3* and *dmd-3* loci were tagged at the C-terminus as previously described (*Dickinson et al., 2015*; *Katic et al., 2015*). The plasmid serving as a template for homologous recombination was cloned using Gibson and included 700 bp-long homology arms. Wild-type moving recombinants were isolated, verified by DNA sequencing and outcrossed three times.

## Neuron identification

Expression analysis of the *lin-29* endogenously tagged alleles with neuron-specific resolution was done by assessing nuclear position and size using Nomarski optics and crossing these strains with neuronal landmark reporter strains *eat-4 (otIs518)* (*Serrano-Saiz et al., 2017a*) and *cho-1 (otIs544)* (*Pereira et al., 2015*).

## Laser ablation

Laser ablation of Z1/Z4 was performed at the L1 stage as previously described (*Hsin and Kenyon, 1999*) using a MicroPoint Laser System Basic Unit (N2 pulsed laser (dye pump), ANDOR Technology) attached to a Zeiss Axio Imager Z2 microscope (Objective EC Plan-Neofluar 63X/1.30 Oil). Worms were recovered by washing with M9 buffer and imaged at the young adult stage. Control animals were treated in the same manner, but not subjected to laser exposure. Gonad and germline removal were confirmed by DIC microscopy.

## Microscopy

Worms were anesthetized using 100 mM of sodium azide and mounted on 5% agarose on glass slides. All images were acquired using a Zeiss confocal microscope (LSM880; Zeiss [Carl Zeiss], Thornwood, NY). Image reconstruction was performed using the ZEN software tool. Maximum intensity projections of representative images are shown. For neurite tracing, confocal Z-stacks were opened using FIJI, and loaded into the Simple Neurite Tracer plugin. All neurites emerging from the soma or posterior process of PDB were traced. The simple neurite tracer plugin was used to analyze the skeletons for neurite length, which were summed to calculate total neurite.

## Mating assays

Animals were synchronized at the L1 stage and grown to the early L4 stage group housed. The day before the mating assay experiment, males at the early L4 stage were separated to a new plate. Male mating behavior was assayed by direct observation of the mating process. A single young adult male was placed with 10–15 L4 *unc-3(e151)* hermaphrodites on a mating plate. The male mating behavior was recorded as previously described (*Euling et al., 1999*). Males were tested for their ability to locate vulva in a mating assay, calculated as location efficiency. The number of passes or hesitations at the vulva until the male firs stops were counted: location efficiency = 1/number of encounters to stop, expressed as percentage.

## Mate-searching assay

Animals were synchronized at the L1 stage and grown to the early L4 stage . The day before the mating assay experiment, hermaphrodites and males at the early L4 stage were separated to a new

plate. The mate-searching assay was performed as previously described (*Lipton et al., 2004*). A single animal was placed in the center of a 15 µl patch of food per 10 cm plate. Each animal that had left the food was scored blindly at 7 time points for a period of 24 hr. A worm was considered a leaver if it was 3 cm away from the food source at the scoring time.

## Automated worm tracking

Automated single worm tracking was performed using the Wormtracker 2.0 system at room temperature (*Yemini et al., 2013*). Animals were synchronized at the L1 stage and grown to the early L4 stage . The day before the tracking experiments, hermaphrodites and males were separated to same-sex group housed plates. Animals were recorded at the young adult stage for 5 min to ensure sufficient sampling of locomotion related behavioral features. All animals were tracked on NGM plates uniformly covered with food (OP50). To avoid potential variability arising due to room conditions, all strains that were compared in a single experiment were recorded simultaneously in identical room condition, along with the controls. Analysis of the tracking videos was performed as previously described (*Yemini et al., 2013*). After correction for multiple testing (Bonferroni correction), features shown in *Figure 8* and *Supplementary file 1* emerged as the ones with most significantly different q-value among the test groups. For the rescue experiment, we only measured these features, permitting us to use the p-value.

## Acknowledgements

We thank Qi Chen for generating transgenic strains, Emily Bayer for generating and providing a *ceh-14* null allele, Bob Horvitz and Hannes Bülow for providing strains and DNA, and members of the Hobert lab for comments on the manuscript. We thank Iskra Katic, Lan Xu, Jun Liu, Magdalene Rausch and Chiara Azzi for their help in generating strains. This work was supported by the NIH (2R37NS039996 to OH), by a Birch/Derchin Fellowship to CW, by the Swiss National Science Foundation (Grant # 31003A_163447 to HG) and by the Novartis Research Foundation through FMI (to HG) and the Howard Hughes Medical Institute (OH).

## Additional information

### Competing interests

Oliver Hobert: Reviewing editor, *eLife*. The other authors declare that no competing interests exist.

### Funding

| Funder | Grant reference number | Author |
|---|---|---|
| Howard Hughes Medical Institute | | Oliver Hobert |
| National Institutes of Health | 2R37NS039996 | Oliver Hobert |
| Schweizerischer Nationalfonds zur Förderung der Wissenschaftlichen Forschung | 31003A_163447 | Helge Großhans |
| Novartis Research Foundation | | Helge Großhans |

The funders had no role in study design, data collection and interpretation, or the decision to submit the work for publication.

### Author contributions

Laura Pereira, Conceptualization, Resources, Formal analysis, Investigation, Visualization, Writing—review and editing; Florian Aeschimann, Resources, Formal analysis, Investigation, Writing—review and editing; Chen Wang, Resources, Formal analysis, Investigation, Methodology, Writing—review and editing; Hannah Lawson, Resources, Formal analysis, Investigation; Esther Serrano-Saiz, Investigation, Acquisition of data; Douglas S Portman, Helge Großhans, Supervision, Funding acquisition,

Writing—review and editing; Oliver Hobert, Conceptualization, Formal analysis, Supervision, Funding acquisition, Writing—original draft, Project administration

### Author ORCIDs

Laura Pereira (ID) https://orcid.org/0000-0002-8239-3703
Florian Aeschimann (ID) http://orcid.org/0000-0001-5213-034X
Helge Großhans (ID) http://orcid.org/0000-0002-8169-6905
Oliver Hobert (ID) http://orcid.org/0000-0002-7634-2854

### Decision letter and Author response

Decision letter https://doi.org/10.7554/eLife.42078.019
Author response https://doi.org/10.7554/eLife.42078.020

## Additional files

### Supplementary files

• Supplementary file 1. List of sexually dimorphic, but *lin-29A* independent locomotory features.
DOI: https://doi.org/10.7554/eLife.42078.016

• Transparent reporting form
DOI: https://doi.org/10.7554/eLife.42078.017

### Data availability

All data generated or analysed during this study are included in the manuscript and supporting files.

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
