## [Decision Letter]

Thank you for submitting your article "Timing mechanism of sexually dimorphic nervous system differentiation" for consideration by *eLife*. Your article has been reviewed by two peer reviewers, and the evaluation has been overseen by a Reviewing Editor and Eve Marder as the Senior Editor. The reviewers have opted to remain anonymous.

Both reviewers were very enthusiastic about the conceptual and technical aspects of the work and recommended acceptance. However, both reviewers asked for very minor edits to the figures. I am listing the suggested revisions below. It should be possible for the Reviewing Editor to check these without requiring rereview.

*Reviewer 1:*

Some of the figure panels are too small or could be cropped differently to show the cells of interest. The images might be readable in the online version but the pdf version has lost resolution for some panels. Some of the cells marked by arrowheads will be nearly impossible to see in print.

Suggested but not required experimental revision:

It would have been nice to see the potential link between UNC-3 and lin-29 tested by removing the conserved binding sites to see if motor neuron expression is affected, but I would not suggest delaying publication for this experiment.

*Reviewer 2:*

- Figure 3: based on the description in the main text, I understood that the lin-29::gfp;lin-29a(xe40) reporter was generated by CRISPR insertion of GFP in a lin-29a(xe40) locus. If this is the case, then, the reporter should not be called lin-29::gfp;lin-29a(xe40) as it appears in the figure but lin-29(xe40)::gfp or lin-29a(xe40)lin-29b::gfp or something along these lines that reflects that the xe40 mutation and GFP insertion are within the same locus.

- Figure 3D: the labelling on the left is incorrect. Should say herm and male (without adult) because the stages are on the top labels.

- Figure 3: for clarity, symbols should always indicate the same. For example, white arrows indicate glia in panel C but neurons in panel D. Also, what do black arrows in D indicate?

- Figure 6C: remove stage (=adult) from the top labels. These should indicate sex only since the stage is indicated in the left axis.

- Final sentence in the Results section: the reference to Figure 9A, B should be to Figure 10A, B.

---

## [Author Response]

Reviewer 1:Some of the figure panels are too small or could be cropped differently to show the cells of interest. The images might be readable in the online version but the pdf version has lost resolution for some panels. Some of the cells marked by arrowheads will be nearly impossible to see in print.

These loss of resolution were a consequence of image size. We are confident that the panels are well readable in the optimal figure format. We have cropped some of the figures but are reluctant to perform cropping of others so as to not lose important information.

Suggested but not required experimental revision:It would have been nice to see the potential link between UNC-3 and lin-29 tested by removing the conserved binding sites to see if motor neuron expression is affected, but I would not suggest delaying publication for this experiment.

We thank the reviewer for suggesting this experiment. We are currently trying to address if we can abolish lin-29 expression in the ventral nerve cord neurons by deleting the endogenous unc-3 binding sites on its promoter. We have so far tested one unc-3 binding site which did not abolish lin-29 expression and we are moving on to testing other sites or testing them in combination.

Reviewer 2:- Figure 3: based on the description in the main text, I understood that the lin-29::gfp;lin-29a(xe40) reporter was generated by CRISPR insertion of GFP in a lin-29a(xe40) locus. If this is the case, then, the reporter should not be called lin-29::gfp;lin-29a(xe40) as it appears in the figure but lin-29(xe40)::gfp or lin-29a(xe40)lin-29b::gfp or something along these lines that reflects that the xe40 mutation and GFP insertion are within the same locus.

Yes, totally right, fixed.

- Figure 3D: the labelling on the left is incorrect. Should say herm and male (without adult) because the stages are on the top labels.

Fixed.

- Figure 3: for clarity, symbols should always indicate the same. For example, white arrows indicate glia in panel C but neurons in panel D. Also, what do black arrows in D indicate?

Fixed.

- Figure 6C: remove stage (=adult) from the top labels. These should indicate sex only since the stage is indicated in the left axis.

Fixed.

- Final sentence in the Results section: the reference to Figure 9A, B should be to Figure 10A, B.

Fixed.